# Injured epithelial cell states impact kidney allograft survival after T-cell-mediated rejection

Anna Maria Pfefferkorn [1,14], Lorenz Jahn [2,14], Patrick T. Gauthier [3,14], Vera Anna Kulow [4], Johannes Roeles[2,5], Niklas Müller-Bötticher[6,7], Louisa M. S. Gerhardt[8], Janna Leiz[2], Sadia Sarfraz[1], Izabela Plumbom [9,10], Robert Greite[2], Svjetlana Lovric [2], Jaba Gamrekelashvili [2], Florian Limbourg[2], Jessica Schmitz [11], Jan Hinrich Bräsen [11], Irina Scheffner [2], Igor M. Sauer [1], Felix Aigner[1,12], Janine Altmüller[9,10], Thomas Conrad [9,10], Wilfried Gwinner [2], Naveed Ishaque[6], Michael Fähling [4], Kai M. Schmidt-Ott [2], Philip F. Halloran[3,13,15] ✉, Muhammad Imtiaz Ashraf [1,15] ✉ & Christian Hinze [2,15] ✉

T-cell–mediated rejection (TCMR) remains a major cause of kidney transplant failure, despite being considered treatable. Its impact reflects a limited understanding of the underlying molecular mechanisms and their clinical consequences. To address this, we induced acute TCMR in mouse kidney transplants and profiled molecular changes using single-nucleus RNA sequencing (snRNA-seq), spatial transcriptomics and immunofluorescence. Results were compared with human snRNA-seq data from TCMR and stable allografts, as well as single-cell deconvolution analysis of bulk transcriptomic data from kidney transplant biopsies. Here we show that TCMR induces injured epithelial cell states in mouse kidney allografts, particularly in proximal tubules and thick ascending limbs. Spatial transcriptomics of these injured epithelial states demonstrated heterogeneous localization, interactions with immune cells and cellular microenvironments. Cross-species analysis confirmed similar severely injured epithelial states in human samples, whose abundances correlated with transplant survival and persisted despite TCMR resolution. Collectively, our results identify epithelial injury cell states as a determinant of outcome after TCMR.

T cell-mediated rejection (TCMR) is a severe complication after kidney transplantation associated with reduced graft survival[1,2]. Often occurring within the first year after kidney transplantation, TCMR represents an adaptive immune response triggered by donor antigens, primarily mediated by T cells[3–6]. This immune reaction causes injury to the transplanted kidney, resulting in inflammation and cellular damage, which can advance to chronic graft dysfunction. TCMR is defined histologically by interstitial leukocyte infiltration, tubulitis (inflammation of renal tubules) and, in some instances, arteritis[5,7,8]. Although the precise mechanisms driving kidney injury during TCMR are not fully understood, they likely involve direct cytotoxic T cell activity, a pro-inflammatory cytokine environment and interstitial edema leading to hypoxia in kidney cells[9–11]. Current therapeutic strategies for TCMR focus on suppressing the immune cell infiltrate through intensified

immunosuppression, often including steroid pulses[12–14]. While these treatments achieve histological remission in most cases, TCMR remains associated with reduced allograft survival, with outcomes at least as poor as those in antibody-mediated rejection (ABMR) despite TCMR being considered treatable[15–18]. This raises questions about whether current therapies adequately address all aspects of injury occurring during TCMR.

Large-scale bulk transcriptomic studies have underscored the importance of epithelial injury signatures, especially in TCMR, noting that these signatures are most relevant for allograft outcomes, whereas immune cell infiltration signatures appear to have limited prognostic value[17,19–21]. Recent advances in high-resolution transcriptomic technologies have offered new perspectives on the molecular mechanisms underlying kidney diseases. These technologies, particularly in the context of acute kidney injury (AKI) and chronic kidney disease (CKD), have revealed the existence of injury-induced epithelial cell states[22–25]. Emerging evidence suggests that these injured cell states are not mere bystanders but actively contribute to further kidney damage through their pro-inflammatory and pro-fibrotic gene expression profiles[23,25,26].

Despite these insights, epithelial injury signatures in TCMR have yet to be systematically characterized or linked to clinical outcomes. The complexity of evaluating these injury states in transplanted human kidneys is compounded by multiple sources of injury, including drug toxicity, ischemia-reperfusion injury as well as host- and donor-specific factors[27–31].

In this work, we delineate the molecular changes associated with TCMR and evaluate their impact on allograft outcomes. We employ single-cell sequencing technologies in mouse models of TCMR to establish a precise signature of TCMR-induced changes. These findings are then rigorously compared to molecular data from human biopsies. By identifying key molecular signatures in TCMR samples and probing them in large bulk transcriptomic cohorts with clinical follow-up data, we can assess their clinical significance.

## Results

### Allogeneic kidney transplantation in mice induces acute TCMR

Analyzing molecular signatures of TCMR in patient kidney biopsies poses significant challenges. Firstly, even when TCMR is present, various overlapping injury sources in human kidney allografts—such as drug toxicity, ischemia-reperfusion injury, diverse donor and recipient pathologies and clinical conditions—are commonly observed[32]. Secondly, the precise timing of TCMR onset in humans is often unknown, complicating inter-patient comparisons.

Mouse models of syngeneic and allogeneic kidney transplantation offer a controlled setting to study TCMR at defined time points[33]. In this study, we therefore propose a cross-species approach by examining gene expression changes in mouse TCMR and systematically comparing them to human TCMR data and clinical outcomes (Fig. 1A).

For the mouse model, we transplanted kidneys from adult male mice of either C57BL/6 or BALB/c strains into syngeneic (C57BL/6 to C57BL/6 or BALB/c to BALB/c, further referred to as syngeneic mice) or allogeneic (C57BL/6 to BALB/c or BALB/c to C57BL/6, further referred to as allogeneic mice) recipients, with a cold ischemia time of 60 minutes. The BALB/c to C57BL/6 allogeneic transplantation was used to model milder rejection compared to the C57BL/6 to BALB/c group, while syngeneic transplants served as controls. Kidneys were harvested 7 days post-transplantation, a time point considered optimal for acute cellular rejection in this model[33]. Allogeneic transplants induced strong rejection, showing all histological features of TCMR, including tubulitis and interstitial inflammation (Fig. 1B, and Suppl. Fig. S1 for Banff scoring, sample information and additional histology). Clinically, allogeneic mice suffered from acute kidney injury (AKI) and increased mortality (Fig. 1B, and Suppl. Fig. S2).

To gain molecular insights into TCMR, we performed single-nucleus RNA sequencing (snRNA-seq) and spatial transcriptomics (ST)

using the Xenium platform (mouse multi-tissue panel plus 100 custom genes; see Suppl. Data 1) on kidneys from syngeneic and allogeneic transplants. Both, snRNA-seq (39706 nuclei from 3 syngeneic kidneys and 5 allogeneic kidneys, Suppl. Fig. S3) and ST (ca. 1.6 million segmented cells from 2 syngeneic and 5 allogeneic kidneys, Suppl. Fig. S4) produced high-quality transcriptomic data, enabling us to identify all expected major cell types. (Fig. 1C and D, Suppl. Fig. S5). Cell types in ST data were obtained by label transfer from snRNA-seq using the singleR package[34,35]. This led to expected marker gene expression (Suppl. Fig. S4A) and spatial localization of cell types (Fig. 1D, and Suppl. Fig S5). In snRNA-seq data, global gene expression was mainly determined by the presence or absence of allogeneic transplantation as derived from principal component analysis (Fig. 1E). Leukocyte abundance was significantly higher in allogeneic samples compared to syngeneic controls in snRNA-seq, correlating with the interstitial inflammation observed in histology (Fig. 1F).

### Mouse TCMR induces pronounced gene expression responses in kidney epithelial cells

Differential gene expression analysis between allogeneic and syngeneic mice revealed the strongest gene expression response in the kidney epithelium, predominantly in proximal tubules (PT) and thick ascending limbs (TAL) (Fig. 2A, and Suppl. Data 2). Epithelial gene expression in TCMR was characterized by signatures of kidney injury (elevated expression of Havcr1 and Lcn2, Fig. 2B, C, Lcn2 particularly pronounced in C57BL/6 to BALB/c kidneys), accompanied by broad upregulation of proinflammatory chemokines (e.g., Cxcl9, Ccl5, Cxcl10) and MHC class I and II molecules (e.g., H2-Q7, H2-Eb1, H2-Q6) across epithelial cell types, consistent with immune pathway activation (Fig. 2D, E), alongside downregulation of genes associated with epithelial differentiation and tubular transport (Suppl. Fig. S6 and Suppl. Data 2). Consistently, pathway enrichment analysis on differentially upregulated genes often showed pan-cell type enrichment in pathways likely associated with the overall inflammatory milieu in the allogeneic kidneys and AKI. This included interferon alpha, interferon gamma, interleukin 2, interleukin 6, TNF alpha and epithelial mesenchymal transition (EMT) signaling (Fig. 2F).

Cell type-specific differentially upregulated genes for PT and TAL closely mirrored this broader spectrum of differentially expressed genes in all other cell types, encompassing interferon-inducible transcripts, chemokines and genes involved in antigen presentation, as well as pathways associated with epithelial stress, injury and damage (Suppl. Fig. S7).

For most cell types, there was a large overlap of differential gene expression between the two different allogeneic models (C57BL/6 to BALB/c or BALB/c to C57BL/6) with a usually stronger gene expression response in C57BL/6 kidneys transplanted into BALB/c mice (Suppl. Fig. S6B). Notably, the number of differentially upregulated genes in some cell types (CD-PC, EC, IntC, PEC, CD-IC-A) was higher in the BALB/c to C57BL/6 kidneys which represents the supposedly milder rejection model (see Suppl. Data 2 for all differentially expressed genes).

### Mouse TCMR elicits spatially diverse injured cell states in PT and TAL

Single-cell studies in AKI have demonstrated the emergence of outcome-relevant AKI-associated cell states within the kidney epithelium[22,23,25]. To investigate whether similar phenomena occur in TCMR, we conducted subclustering analyses of the snRNA-seq data from PT and TAL cells, which exhibited the strongest gene expression changes.

In the PT, four injury clusters (PT Injury m1-4) and a cluster of proliferating cells (PT Prolif) were identified (Fig. 3A-D). PT Injury m1 cells showed early epithelial stress adaptation with increased expression of Cryab, Myo5b, Serpina10, Acsm3 and Stk39. Concurrent upregulation of Jag1, Wnt5a and Foxc1 may indicate engagement of

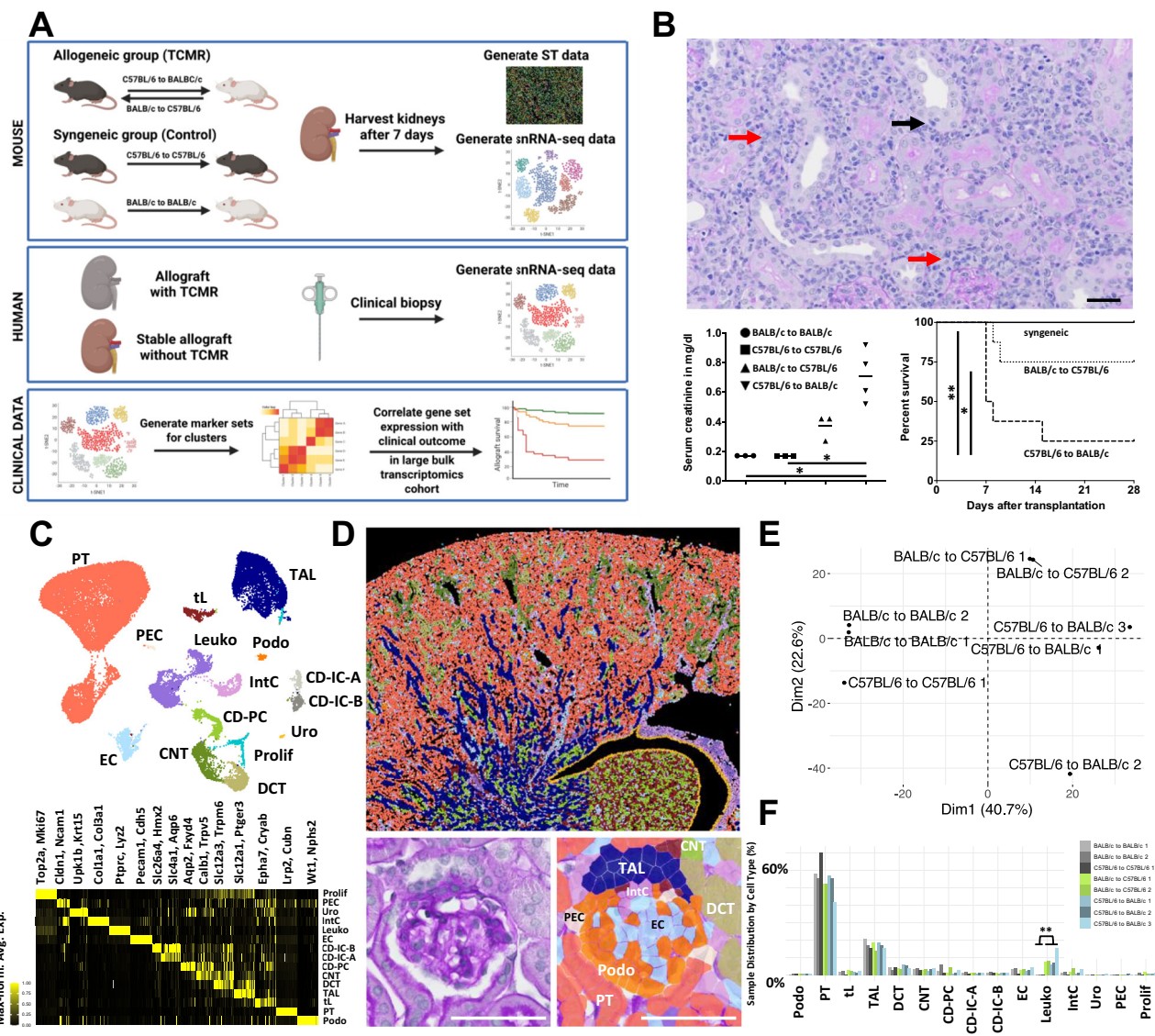

**Fig. 1 | Mouse allogeneic kidney transplantation induces TCMR and is associated with increased mortality and occurrence of AKI. A** Schematic overview depicting the overall study design including mouse and human samples. Created in BioRender. Hinze, C. (https://BioRender.com/0rgf9mf). **B** Upper panel: Representative periodic acid-Schiff staining of a C57BL/6 kidney transplanted into a BALB/c mouse 7 days after transplantation showing interstitial inflammation (red arrows) and tubulitis (black arrow). Scale bar: 50 μm. Lower left panel: Significantly elevated serum creatinine levels in allogeneic kidneys ($n = 3$ BALB/c to BALB/c, C57BL/6 to C57BL/6 and BALB/c to C57BL/6; $n = 4$ C57BL/6 to BALB/c, Kruskal-Wallis test followed by Dunn's test for multiple comparisons using GraphPad Prism 8.0.1). The mean value of each group is presented. Lower right panel: Kaplan-Meier survival curves of the transplanted mice, indicating significantly reduced survival of the allografts compared to the isografts ($n = 7$ C57BL/6 to C57BL/6, n = 8 BALB/c to C57BL/6, C57BL/6 to BALB/c, Mantel-Cox log-rank and Gehan-Breslow-Wilcoxon tests for group comparison using GraphPad Prism 8.0.1. **C** Uniform manifold approximation and projection (UMAP) of snRNA-seq from mouse kidney

transplants highlighting all major cell types and heatmap depicting marker gene expression. Selected markers are highlighted above the heatmap. **D** ST data of a syngeneic kidney (C57BL/6 to C57BL/6 7 days after transplantation) showing the spatial distribution of major cell types (repeated seven times with similar result). Cell type colors are identical to those used in the UMAP. Scale bar: 500 μm. The lower panels show a magnification of a glomerulus from the same kidney with the periodic acid-Schiff staining on the left and cell type identities from ST data overlayed on the right. Scale bar: 50 μm. **E** Principal component analysis on PT pseudobulk data from snRNA-seq and PT-specific highly variable genes. **F** Relative abundances of major cell types in snRNA-seq samples. Two-sided Student's t-test with Benjamini-Hochberg correction for multiple testing. *P* value: *<0.05, **<0.005. Prolif - proliferation, *PEC* parietal epithelial cells, *Uro* urothelium, *IntC* interstitial cells, *Leuko* leukocytes, *EC* endothelial cells, CD-PC/IC-A/IC-B – collecting duct principal/type A and type B intercalated cells, *CNT* connecting tubule, *DCT* distal convoluted tubule, *TAL* thick ascending limb, *tL* thin limb, *PT* proximal tubule, *Podo* podocyte. Source data are provided as a Source Data file.

developmental or regenerative transcriptional programs (Fig. 3C). PT Injury m2 exhibited elevated expression of MAPK pathway components (*Map3k1*, *Map3k13*) and stress-responsive genes including *Egfr* (Fig. 3C). Cells in the PT Injury m3 cluster expressed genes associated with complement and coagulation pathways (*C3*, *Pros1*, *Sparc*), along with immune and interferon-related transcripts such as *Cd74*, *Stat1*, *B2m* and *Klf7* (Fig. 3C). The most dedifferentiated injury state (as determined by pseudotime analysis, Fig. 3A), PT Injury m4, exhibited a

broad injury-associated transcriptional program. This included expression of injury markers *Cd44* and *Havcr1*, pro-inflammatory mediators (*Cxcl10*, *Nfkb1*, *Atf3*) and genes associated with epithelial plasticity and epithelial–mesenchymal transition, such as *Vim*, *Itgb1*, *Tnfaip3*, *Tpm1* and *Vcam1*. Markers such as *Ccl2*, *Cd47*, *Icam1* and *Anxa3* may indicate epithelial signals promoting immune cell recruitment and interaction (Fig. 3C). Many of these markers including *Ccl2*, *Vim* and *Vcam1* are reported to be involved in maladaptive repair after AKI[22,24].

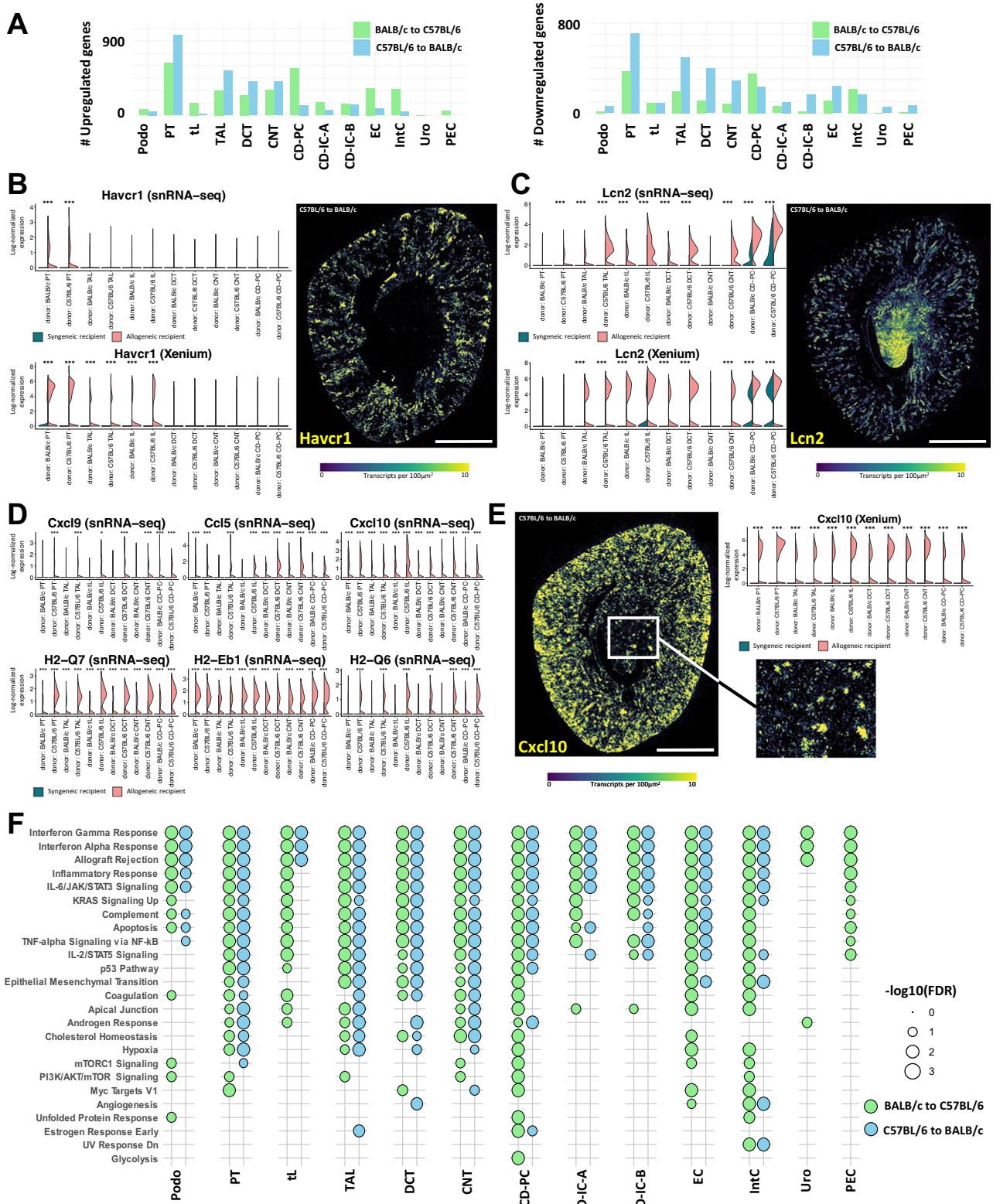

**Fig. 2 | Mouse allogeneic transplantation induces strong epithelial gene expression response. A** Bar plots showing the number of differentially upregulated (left) and downregulated (right) genes across major kidney cell types for BALB/c to C57BL/6 (green) and C57BL/6 to BALB/c (blue) transplants when compared to the respective syngeneic control. **B** and **C** Violin plots in mouse snRNA-seq and Xenium data as well as transcript density plots (bin size 10 μm) for known injury genes *Havcr1* and *Lcn2*. **D** Top upregulated pro-inflammatory cytokines and genes encoding for mouse MHC proteins in epithelial cells in snRNA-seq data. **E** Xenium data for differentially upregulated gene *Cxcl10* shows strong upregulation in cortical areas but also more patchy regions in the medulla (highlighted in white box).

**F** Dot plot of pathway enrichment analysis of differentially upregulated genes for each major cell type and transplant group, e.g., BALB/c to C57BL/6 (green) and C57BL/6 to BALB/c (blue). Shown are all significant pathways (FDR < 0.05) from gene set enrichment analysis using hallmark gene sets. Scale bar **B**, **C**, **E**: 1000 μm. *P* values were calculated in (**B**–**E**) using a two-sided Wilcoxon rank sum test (Seurat FindMarkers function) and adjusted for multiple testing with the Benjamini–Hochberg procedure. Adjusted *P* value: *<0.05, ***<0.001. Significances are shown for cell types with 25% expressing cells in allogeneic model. Source data are provided as a Source Data file.

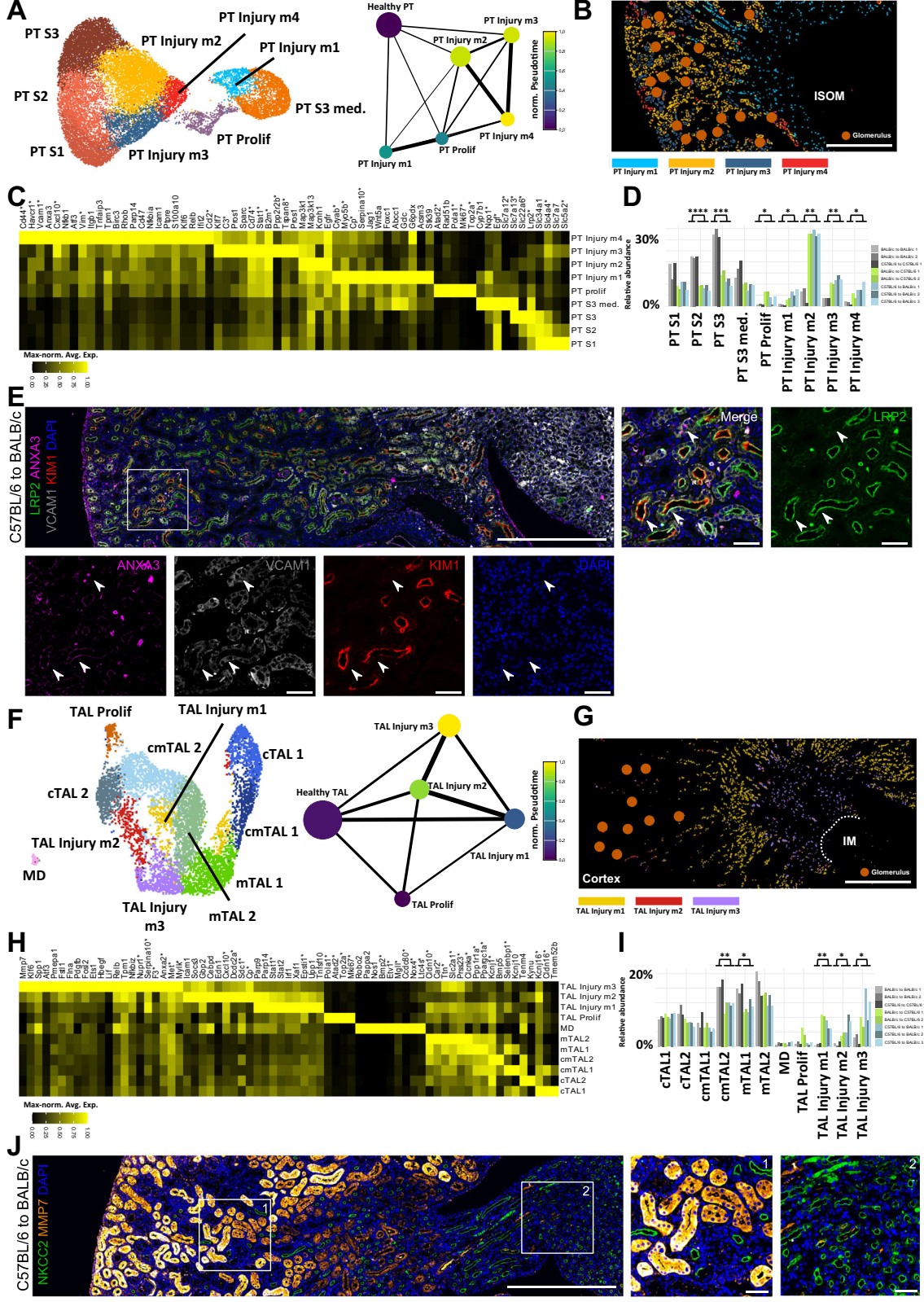

Label transfer from snRNA-seq to Xenium spatial transcriptomics revealed a healthy PT cell distribution consistent with known anatomy (Supplementary Fig. S8). Marker gene expression showed strong concordance between platforms, confirming cross-platform alignment (Fig. 3C and Supplementary Fig. S8B). Spatial mapping of injury-associated PT states (Fig. 3B) demonstrated that PT Injury m1 localized primarily to the outer medulla and

medullary rays, as confirmed by Xenium data and immunofluorescence staining for *ACSM3* and *STK39* (Supplementary Figs. S9 and S10). In contrast, PT Injury m4 exhibited a patchy distribution across cortex and medulla, aligning with the spatial expression domains of *Havcr1* and *Vcam1*, which were validated by Xenium data and immunofluorescence staining (Fig. 3E, and Supplementary Figs. S9 and S10).

**Fig. 3 | TCMR induces spatially diverse injury cell populations in PT and TAL.**
**A** SnRNA-seq UMAP plot (left) of PT subclustering including anatomical segments (PT S1-3, PT S3 med.), a proliferative PT cluster (PT Prolif) and injured clusters (PT Injury m1-4). Pseudotime analysis using partition-based graph abstraction (PAGA) highlighting diffusion pseudotime (right). **B** Spatial localization of PT injury clusters derived from snRNA-seq label transfer to ST data. **C** Heatmap with expression of marker genes across PT subclusters, maximum-normalized on a per-gene basis. Genes present in the Xenium panel are marked with an asterisk (compare to Suppl. Fig. S8). **D** Relative abundances of PT subclusters across different transplant groups. **E** Immunofluorescence staining of PT Injury m4 marker genes VCAM1, ANXA3, KIM1 and canonical PT marker LRP2 in a C57BL/6 to BALB/c kidney (see Suppl. Fig. S9 for remaining groups). Consistent with snRNA-seq predictions, VCAM1, KIM1 and ANXA3 share overlapping expression domains that diminish in size from VCAM1 to KIM1 to ANXA3. Nuclei are counterstained with DAPI. Experiment repeated three times with similar result. **F−I** Analogous plots for the TAL. **J** Immunofluorescence staining validates the TAL Injury m3 marker MMP7, detected in scattered medullary NKCC2-positive TAL cells in allogeneic but not syngeneic grafts. Nuclei are counterstained with DAPI. MMP7 also labels PT cells. Remaining groups are shown in Supplementary Fig. S12. Experiment repeated three times with similar result. **D**, **I** show results from two-sided Student's t-test with Benjamini-Hochberg correction for multiple testing. Adjusted *P*-value: *<0.05, **<0.005 (for **D**, **I**). Scale bar (**B**, **G**): 500 μm, scale bar large images (**E**, **J**): 1000 μm, scale bar small images (**E**, **J**): 50 μm. Source data are provided as a Source Data file.

In the TAL, three TCMR-associated injury clusters (TAL Injury m1-3) and a proliferating cluster (TAL Prolif) could be observed (Fig. 3F-I). The TAL Injury m1 cluster showed a modest injury profile characterized by activation of interferon-stimulated genes and immune regulators, including *Parp9, Parp14, Stat1, Stat2, Irf1, Xaf1, Epsti1* and *Upp1* (Fig. 3H). TAL Injury m2 markers included genes involved in pro-inflammatory and interferon-driven transcriptional programs such as *Icam1, Socs3, Gbp2, Cebpd, Edn1* and *Cxcl10*. Marker genes such as *Dcdc2a* and *Sdc1* suggest subtle changes in cellular organization and epithelial remodeling (Fig. 3H). By contrast, TAL Injury m3 exhibited a broad and complex injury signature. Core injury markers such as *Mmp7, Spp1* and *Atf3* are co-expressed with EMT- and cytoskeleton-associated genes such as *Pmepa1, Fstl1, Flna* and *Pdgfb*, reflecting epithelial plasticity and tissue remodeling (Fig. 3H). As PT Injury m4, TAL Injury m3 exhibited similarities to severely injured TAL cells in AKI, which we identified in a previous study[22].

Healthy TAL cells showed an expected spatial localization and marker gene expression was consistent across snRNA-seq and Xenium datasets (Fig. 3H and Supplementary Fig. S11). Among the injury-associated TAL states, TAL Injury m2 localized to the cortex, TAL Injury m1 to the medulla and the most severely injured population, TAL Injury m3 (as defined by pseudotime analysis, Fig. 3F), was predominantly restricted to the inner stripe of the outer medulla, with few occasional cells also detected in medullary rays (Fig. 3G). The presence and location of TAL Injury m3 cells was validated by immunofluorescence staining for the marker gene *Mmp7* (Fig. 3J and Supplementary Fig. S12).

It is of note that subclustering analyses of the remaining major cell types revealed TCMR-associated injury clusters in the distal convoluted tubule (DCT), connecting tubule (CNT) and endothelial cells (EC) at much lower abundances than for PT and TAL (Suppl. Figs. S13 and S14).

## TCMR-induced injured epithelial cell states show heterogeneous proximity to immune cells and surrounding cell type environments

Although the exact mechanisms driving epithelial injury in TCMR remain unclear, it is widely recognized that host leukocytes play a central role, either by direct cytotoxicity or indirectly by cytokine production. The availability of snRNA-seq and ST data allows for estimating the spatial proximity of leukocyte subtypes to injured epithelial cells in the PT and TAL. Subclustering of leukocytes identified all expected subtypes, most of them significantly overrepresented in allogeneic kidneys (Fig. 4A−C). The majority of leukocytes consisted of macrophage subtypes and CD8+ T cells as confirmed by the expression of hallmark genes (Fig. 4A, C). Spatial analysis showed that most leukocytes were located in the cortex and around blood vessels (Fig. 4D). However, both, T cells and macrophages were also observed in the inner medulla.

For spatial proximity analysis, we analyzed the co-localization of leukocytes in the direct neighborhood (defined by ≤ 25 μm distance between cells) of injured cells from PT and TAL (Fig. 4E). The highest probability of directly neighboring leukocytes could be observed in the injured PT cell states and was particularly high in PT Injury m3 and m4 (Fig. 4E). The most severely injured TAL cell state, TAL Injury m3 and also TAL Injury m1 showed the strongest depletion of neighboring leukocytes. These results show a differential interaction of leukocytes with different injured cell states in PT and TAL during TCMR. This general pattern was unaffected by the distance threshold used to define direct neighbors (Suppl. Fig. S15) and persisted when assessing the spatial proximity of injured PT and TAL cell states to leukocyte subtypes (Suppl. Figs. S16 and S17). It is of note that proximity of activated/ lipid-associated macrophages with injured PT cells was particularly high (Suppl. Fig. S16).

To further characterize the cellular microenvironments in our mouse model, we performed unbiased spatial domain identification using the SpatialLeiden package[36] (Fig. 5A-D). Coarse clustering revealed anatomically meaningful and distinct domains, including glomeruli (cluster 8), two medullary compartments (clusters 2 and 3) and several cortical clusters (remaining clusters) (Fig. 5A, B). To examine the microenvironments surrounding PT Injury m4 and TAL Injury m3, we subclustered cortical clusters 0, 1, 6 and medullary cluster 2, separately (Fig. 5C, D). In the cortex, three microdomains (clusters 0, 3, 4) were enriched in allogeneic kidneys (Fig. 5C). Clusters 0 and 4 consisted almost exclusively of endothelial, immune and interstitial cells, whereas cluster 3 contained a high proportion of injured PT cells (PT Injury m2−m4) alongside T cells, macrophages and interstitial cells, suggesting a potentially pro-inflammatory and pro-fibrotic niche. In the medulla, a single cluster enriched in allogeneic kidneys (cluster 0) comprised mainly immune and interstitial cells with only a small fraction of injured TAL cells and a large abundance of intact epithelium. Notably, TAL Injury m3 cells clustered with immune and interstitial cells, though less extensively than the patchy arrangement characteristic of PT Injury m4 and were more diffusely distributed (see also Fig. 3E, J). Immunofluorescence staining validated the spatial proximity of macrophages, T cells and fibroblasts with PT Injury m4 cells (defined by KIM1 expression) (Fig. 5E, and Suppl. Fig. S18 for TAL Injury m3).

## Human TCMR kidney allografts exhibit injured cell states resembling those in mice

To compare the mouse TCMR data with human samples, we performed snRNA-seq on three TCMR and three stable allograft biopsies from archived samples of the protocol biopsy program at Hannover Medical School (see Suppl. Table S1 for patient and biopsy details). The snRNA-seq generated high-quality transcriptomic data, with 22183 nuclei used for analysis, allowing the identification of all major cell types (Fig. 6A, B, and Suppl. Fig. S19). Unlike the mouse data, we did not observe a distinct cluster of proliferating cells, likely due to the earlier time point of TCMR injury in the mouse samples.

Leukocyte subclustering revealed similar cell types as in mouse data (Suppl. Fig. S20). Subclustering of the PT and TAL revealed the presence of injured PT and TAL states (Fig. 6C−H), characterized by reduced expression or absence of canonical markers and upregulation

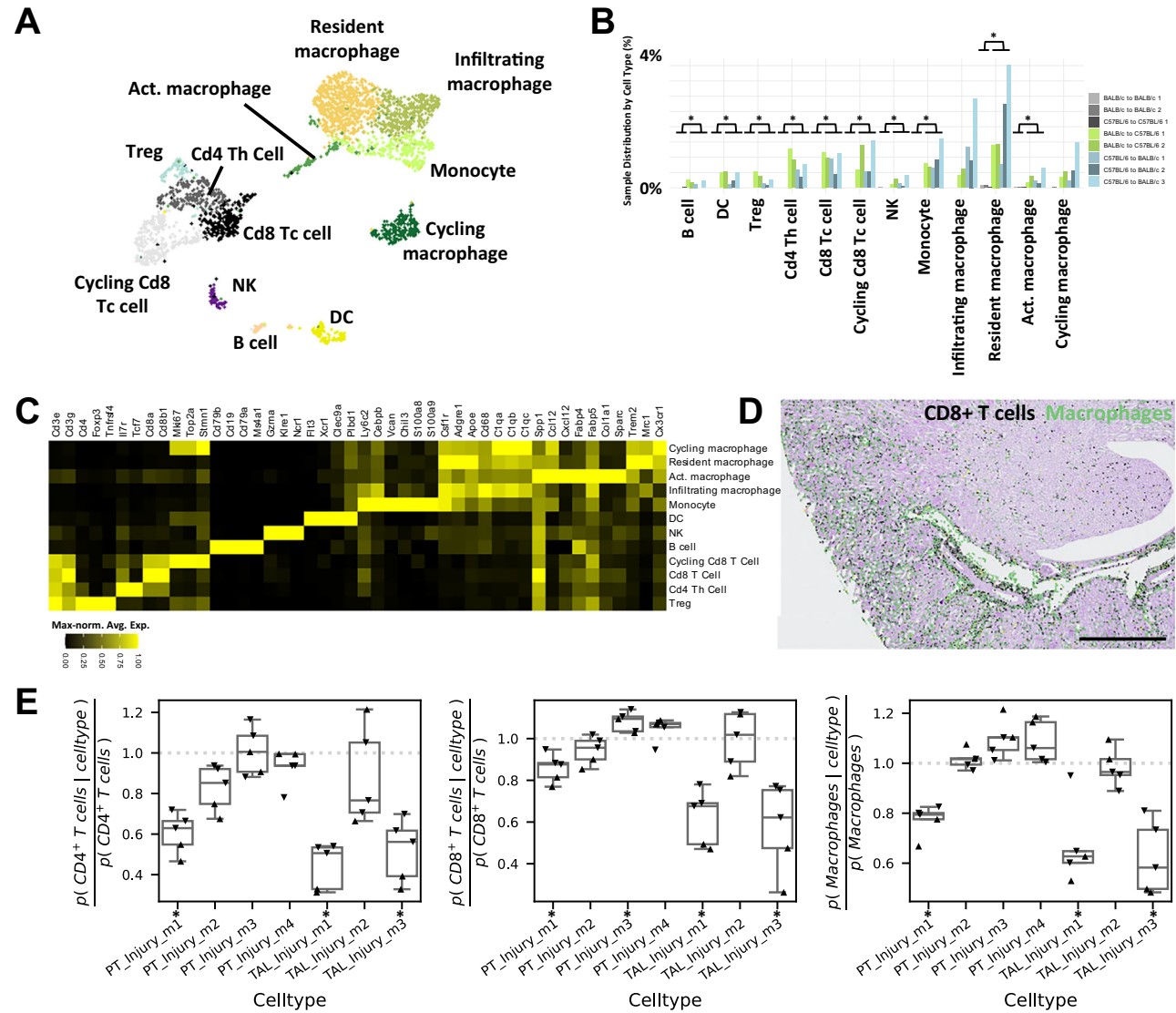

**Fig. 4 | Leukocyte cell types exhibit varying proximities to TCMR-induced injured epithelial cell states within PT and TAL. A** SnRNA-seq UMAP plot of subclustering analysis of all mouse leukocytes. **B** Relative abundances of leukocyte subtypes in individual samples. Two-sided Student's t-test with Benjamini-Hochberg correction for multiple testing. Adjusted *P* value: *<0.05. **C** Marker gene heatmap of identified leukocyte cell types. Gene expression was maximum-normalized per gene. **D** Spatial distribution of CD8⁺ cells and macrophages in C57BL/6 to BALB/c ST data. Notably, the remaining broad leukocyte cell types are not shown and would be not visible due to the magnification and their low abundances. Scale bar: 500 µm. **E** Co-occurrence analysis for PT and TAL injury clusters directly neighboring a broad leukocyte cell type. Allogeneic transplantation C57BL/6 to BALB/c (triangles pointing downwards, n = 3) and BALB/c to C57BL/6 (triangles pointing upwards, n = 2). Box plots show the median and the 25th–75th percentiles, with whiskers extending to the most extreme data points within 1.5× the interquartile range. Two-sided one sample t-test with Benjamini-Hochberg correction for multiple testing. Adjusted *P* value: *<0.05. Significances are depicted next to the x axis labels. Source data are provided as a Source Data file.

of injury markers, similar to those observed in the mouse model (Fig. 6C, F). However, the relative abundances of injured human PT and TAL cells was not significantly higher in TCMR samples when compared to stable allografts. This aligns with the broader spectrum of injury sources in human kidney allografts beyond TCMR, the variable timing of biopsy collection relative to TCMR onset and the limited number of samples in our human snRNA-seq data. To overcome these limitations, we performed single-cell deconvolution of microarray data from Molecular Microscope Diagnostic System (MMDx) signouts, comprising 3858 biopsies from 3210 patients (2155 without rejection, 1107 with antibody-mediated rejection, 298 with TCMR and 298 with mixed rejection), as previously described by us[37]. As expected, these analyses revealed a significant depletion of healthy PT and TAL cell types in TCMR, mixed rejections and ABMR (Fig. 6I). We could also observe a significant enrichment of injury clusters PT Injury h2 and TAL

Injury h2 in TCMR and mixed rejection when compared to no rejection and to ABMR (Fig. 6I). PT Injury h1 and TAL Injury h1 were depleted in TCMR, mixed rejection and ABMR. Microarray data confirmed that gene expression profiles of TCMR in protocol biopsies (used for our human snRNA-seq data) were highly similar to those in indication biopsies (Suppl. Fig. S21A).

To bridge the mouse and human datasets, we performed a cross-species analysis of PT and TAL injury states using label transfer from mouse to human cell types with singleR. As suggested from the marker gene expression (Fig. 6C, F), the most severely injured mouse PT and TAL cell states (PT Injury m4 and TAL Injury m3) corresponded to human PT Injury h2 and TAL Injury h2, respectively (Fig. 6E, H). Human PT Injury h2 expressed all markers of PT maladaptive repair, similar to PT Injury m4, including *VCAM1*, *CD44* and *VIM* (Fig. 6C). The TAL Injury h2 cluster showed a pronounced EMT signature with markers such as

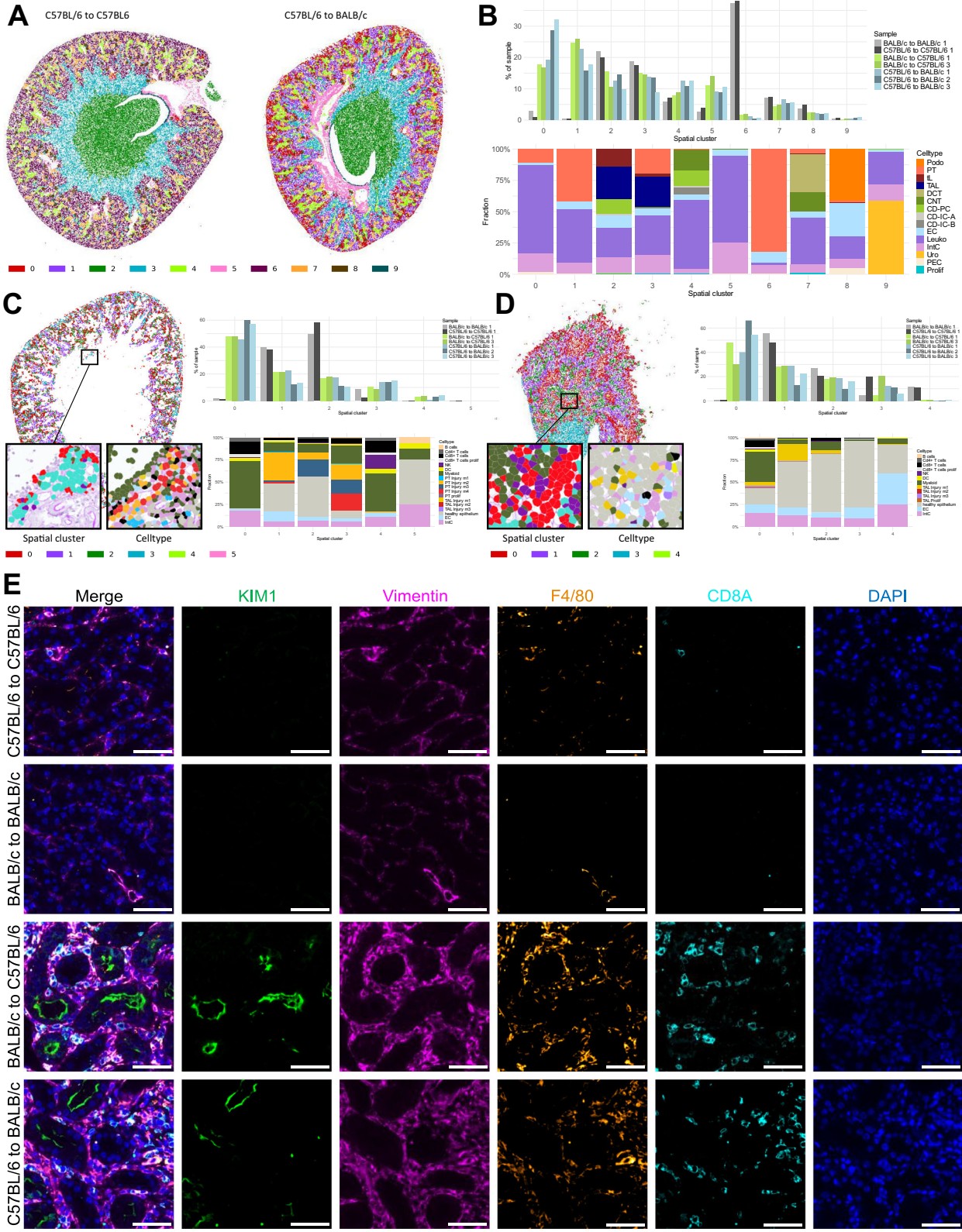

**Fig. 5 | Spatial domain analysis reveals niches enriched in injured PT cells, interstitial cells, T cells and macrophages. A** Coarse spatial domains identified by the SpatialLeiden package on syngeneic and allogeneic kidneys. **B** Abundances and broad cell type composition of spatial domains. **C** Spatial subclustering of coarse clusters 0, 1 and 6 is shown with plots analogous to (**A**) and **B** including a zoom-in of a cluster 3 microdomain containing injured PT cells (PT Injury m2–4), immune cells and interstitial cells. Cell type coloring in the zoom-in matches that in the bar plots. **D** Analogous plots for subclustering of medullary coarse cluster 2 from (**A**). **E** Co-staining for F4/80 (macrophages, orange), CD8A (CD8⁺ T cells, cyan), Vimentin (fibroblasts and macrophages, magenta) and PT Injury m4 marker gene KIM1 (green). Nuclei are counterstained with DAPI. Scale bar: 50 μm. Experiment repeated three times with similar result. Source data are provided as a Source Data file.

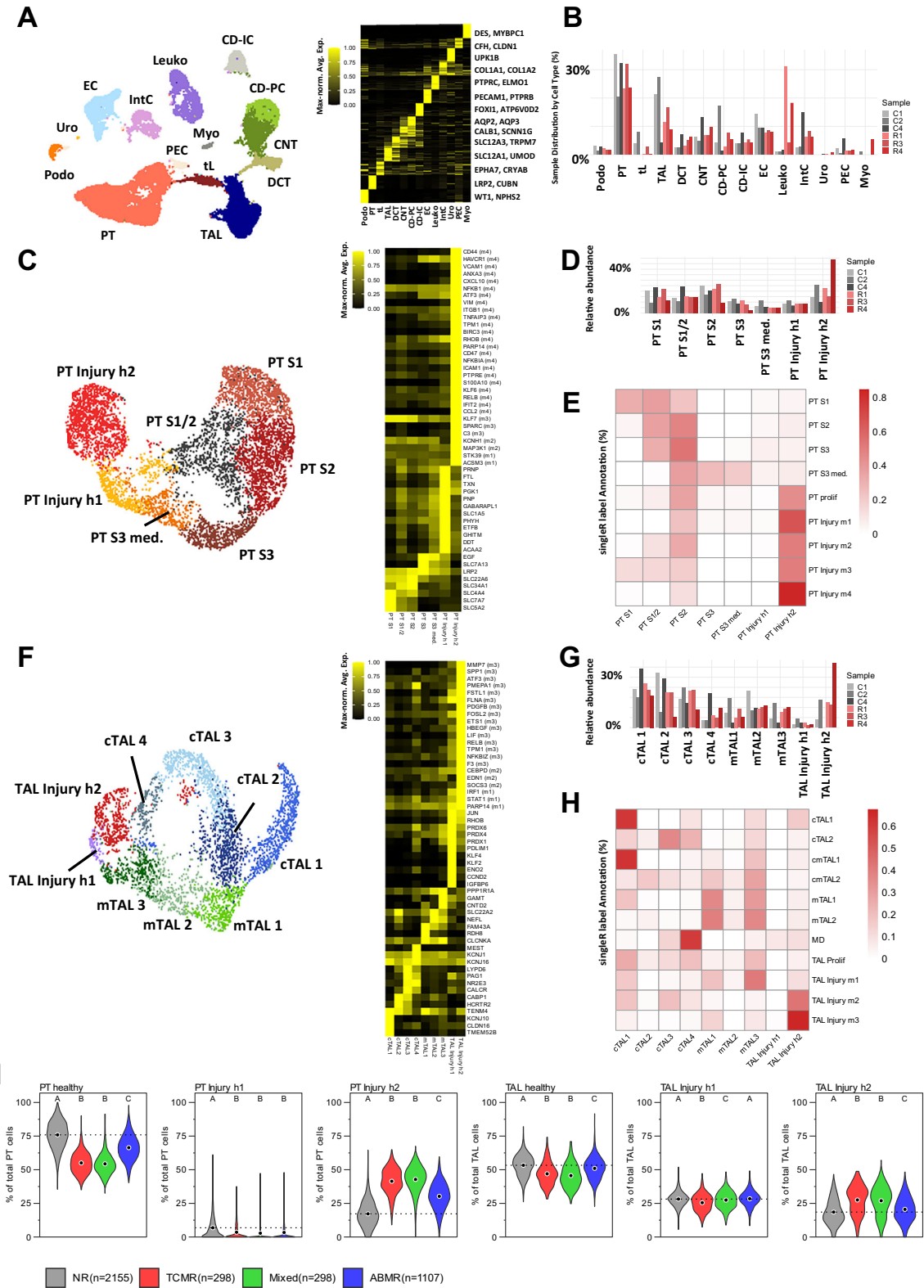

*MMP7, TPM* and *SPP1*, associated with severe TAL injury in AKI and marker genes of TAL Injury m3.

We also leveraged microarray data from 3858 MMDx biopsies (the same population used in our single-cell deconvolution analysis) to systematically compare TCMR gene expression with our mouse and human snRNA-seq datasets, revealing significant correlations (Suppl. Fig. S21B, C).

We validated marker genes of PT Injury h2 and TAL Injury h2 shared with their mouse counterparts using immunofluorescence staining (Fig. 7A, B, compare to Fig. 3, Suppl. Figs. S9, S10 and S12). The data further confirmed that T cells, macrophages and interstitial cells frequently located around PT Injury h2, whereas such associations were much sparser around TAL Injury h2 (Fig. 7C and Suppl. Fig. S22), consistent with mouse kidney stainings (Fig. 5 and Suppl. Fig. S18). For

**Fig. 6 | Most severely injured mouse PT and TAL clusters correspond to injury clusters in human kidney allografts.** **A** UMAP plot of kidney biopsies from human TCMR samples and stable allografts (*n* = 3) displaying major cell types and heatmap with marker gene expression. Selected marker genes are plotted right of the heatmap. **B** Relative abundances of major cell types in individual samples. **C** Subclustering of human PT cells with corresponding marker gene heatmap. Markers also occurring in mouse PT subclustering are highlighted in brackets after the respective gene. **D** Relative abundances of PT subclusters per sample. **E** Classification of human PT subclusters using mouse PT subclusters and singleR label transfer. **F**–**H** Analogous plots for TAL. **I** Comparison of single-cell

deconvolution results for PT and TAL cells from human bulk microarray data in no rejection (NR, *n* = 2155), TCMR (*n* = 298), ABMR (*n* = 1107) and mixed rejection (both TCMR and ABMR, *n* = 298). Solid circles represent the mean % for each rejection group in the respective cell type (PT or TAL). Dotted horizontal lines denote the mean % in NR for visual aid in comparing mean values across groups. Statistical comparisons are denoted by a compact lettering display above each violin. Groups that share the same letters are not significantly different from one another. Differences in % composition among rejection groups were assessed using two-sided arcsin-transformed Bayes-moderated t-tests. Source data are provided as a Source Data file.

immunofluorescence validation, we selected the mouse TAL Injury m3 marker gene *MMP7* as a marker for TAL Injury h2. For PT Injury h2 we chose *KIM1*, *ANXA3* and *VCAM1* (all PT Injury m4 markers) as well as *STK39* and *ACSM3* (PT Injury m1 markers) (Suppl. Fig. S23). Although grouped within a single human snRNA-seq cluster, we observed mostly overlapping expression domains for KIM1 and VCAM1 that differed somewhat from those of STK39 and ACSM3, consistent with mouse data. This suggests potential heterogeneity within PT Injury h2 not fully resolved by clustering.

### Injured epithelial cell states significantly impact kidney allograft survival after TCMR

To assess the impact of injured human PT and TAL cell states, we generated biomarker gene sets which were highly specific for PT Injury h1 and h2 and TAL Injury h1 and h2 (Suppl. Fig. S24 and S25, Suppl. Data 3). To correlate these signatures with clinical outcome, we investigated their correlation with 3-year kidney allograft outcome in bulk transcriptomic data from a large kidney transplant bulk transcriptomics cohort comprising 1061 patients (including 624 with no rejection, 297 antibody-mediated rejections, 95 TCMRs, 45 mixed rejections) (Fig. 8). Scores for provided marker gene sets were derived from geometric mean expression of all marker gene sets in bulk data. We additionally included the IRRAT30 gene set representing injury-induced transcripts which were previously reported to be an important factor of kidney allograft survival after TCMR[17].

For TCMR, higher scores for PT Injury h2, TAL Injury h1 and TAL Injury h2 were significantly associated with an increased risk of allograft loss within 3 years post-biopsy (Fig. 8A). Conversely, high scores for PT Injury h1 (not specific for TCMR) were associated with improved allograft survival after 3 years. This suggests that PT Injury h1, which we interpret as a less severely injured state, may represent a potentially regenerating PT cell phenotype. Similar correlations were observed for ABMR and mixed rejection (Suppl. Fig. S26).

In an exploratory analysis, we assessed whether gene set scores persist over time in patients initially diagnosed with TCMR. This analysis was limited to 12 cases with TCMR that had available follow-up biopsies with resolution to a no-rejection diagnosis in the final biopsy (Fig. 8B). Despite the limited number of patients in this analysis, two observations emerged. First, the levels of injury-related gene set scores exhibited large variability among TCMR patients. This aligns with our findings in AKI, where heterogeneous distributions of injured epithelial cell states were observed among patients[22]. Second, for PT Injury h2 and TAL injury states in particular, a subset of patients demonstrated persistently high scores specific to these injured epithelial cell states, even after the resolution of the initial TCMR diagnosis. This suggests that, while kidneys may achieve remission from TCMR, they can still harbor elevated levels of outcome-relevant injured epithelial cells months after the initial episode.

It is of note that the gene sets specific for PT and TAL Injury h1 and h2 did also successfully label injured PT and TAL cells in AKI from our previous study (Suppl. Fig. S27)[22]. While the distinction between the different injured PT and TAL states was not as pronounced as for TCMR, we still observed correspondence of TAL Injury h1 and h2 with severely injured TAL cells from AKI (TAL-New 3 and 4). PT Injury h1

corresponded mostly to a less severely injured PT cell state (PT-New 1) which expresses genes involved in oxidative stress. PT Injury h2 was strongly associated with severe and potentially maladaptive PT injury in AKI (PT-New 4).

## Discussion

In this study, we explored the molecular changes associated with TCMR and their clinical impact on kidney allograft survival. Using mouse models of TCMR, we identified molecular changes, including a significant gene expression response in most kidney cell types, with the most pronounced effects in the PT and TAL. TCMR was linked to the emergence of injured epithelial cell states, primarily in PT and TAL. Cross-species analysis enabled us to correlate these injured cell states in mice with corresponding states in human biopsies from TCMR patients and stable allograft controls. Single-cell deconvolution and cross-species analyses showed that severely injured PT and TAL cell states (PT Injury h2 and TAL Injury h2) were overrepresented in human TCMR and corresponded to the most severely injured mouse cell states in these cell types (PT Injury m4 and TAL Injury m3). By deriving biomarker gene sets specific to each human injury cell state in PT and TAL, we calculated a score for these gene sets in a large kidney transplant bulk transcriptomics cohort and correlated these scores with clinical outcomes. We found that all identified injury clusters in humans were associated with allograft survival, with all but one gene set score (PT Injury h1) linked to reduced allograft survival. As detailed clinical covariates and formal controls for cell composition were not available for multivariable modeling, our bulk transcriptomic associations cannot be fully evaluated for their independent prognostic effects.

The clinically most relevant cell states with respect to reduced allograft outcome, PT Injury h2, TAL Injury h1 and h2, were characterized by marker genes previously identified in various kidney disease contexts beyond transplantation[22–25,32,38]. Similar cell states have been observed in AKI and CKD, in, both, mouse and human samples[22,24]. Notably, *VCAM1*-positive PT cells, often referred to as maladaptive or "failed repair" cells, reflect significant dedifferentiation and exhibit pro-inflammatory and pro-fibrotic phenotypes. Injured TAL cell states, including the EMT-upregulated genes in TAL Injury h2 (e.g., *MMP7*, *SPP1*, *TPM1*), have been reported primarily in human AKI and CKD studies[22,23]. In non-transplant settings, such PT and TAL cell states have been associated with reduced clinical outcomes[23].

It is noteworthy that although most marker genes of human PT Injury h2 were shared with mouse PT Injury m4 (Fig. 6C), consistent with the strongest correspondence between these clusters in the cross-species analysis (Fig. 6E), PT Injury h2 also contained marker genes characteristic of other mouse PT injury clusters, including PT Injury m1. This observation may point to an unresolved heterogeneity within the PT Injury h2 cluster, potentially due to the limited number of available human snRNA-seq samples.

Among the human TAL injury clusters, TAL Injury h1 displayed a paradoxical pattern of being associated with poorer graft outcome yet relatively enriched in biopsies without rejection and lacking a clear mouse counterpart. TAL Injury h1 may therefore reflect a stress- or recovery-associated TAL program independent of alloimmune injury,

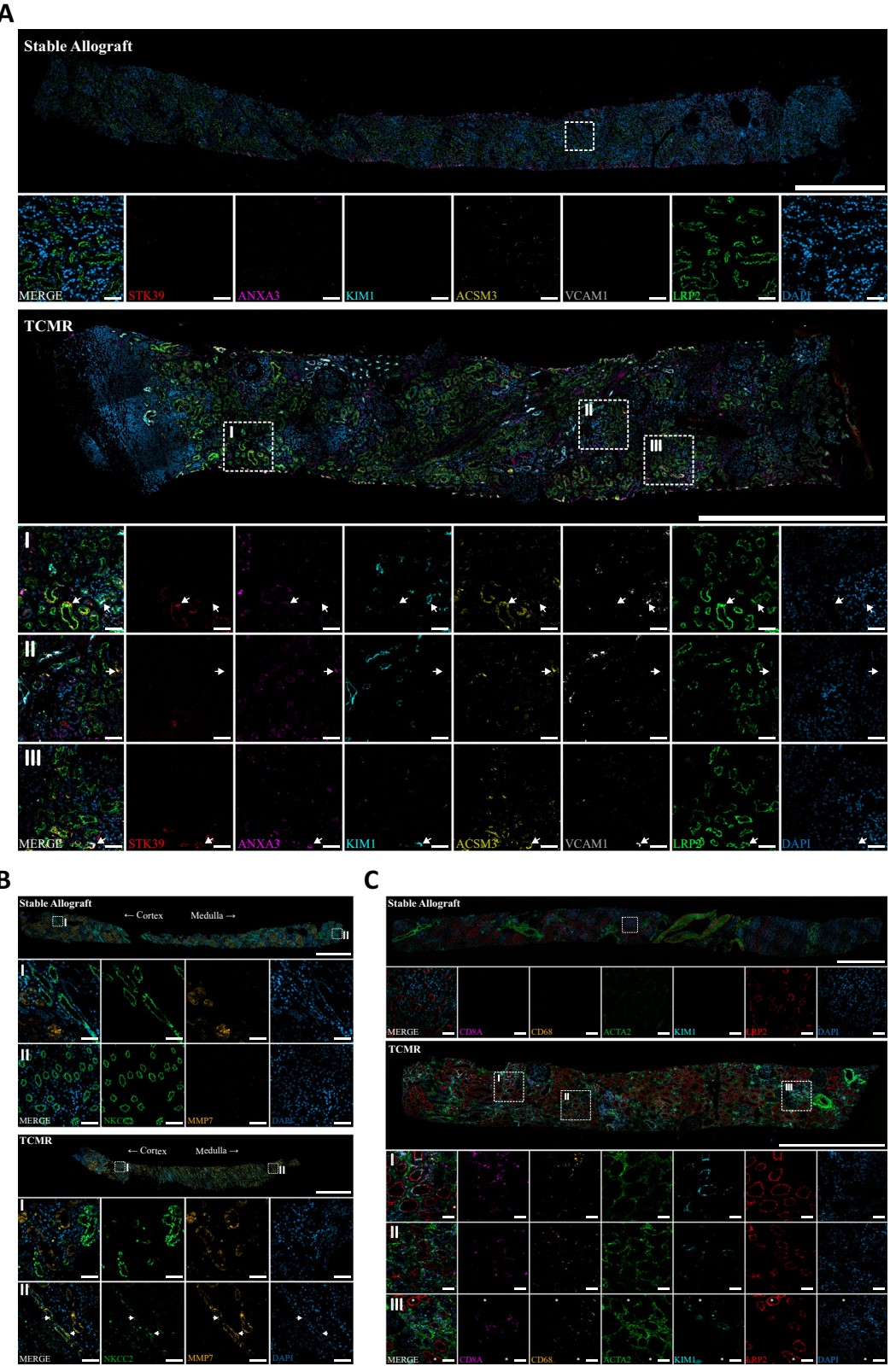

**Fig. 7 | Immunofluorescence staining of kidney biopsy sections from patients with stable allograft function and patients with TCMR. A** Representative overview and inset images of proximal tubule injury showing STK39 (red), ANXA3 (magenta), KIM1 (cyan), ACSM3 (yellow), VCAM1 (gray), LRP2 (green) and DAPI (blue). Arrowheads mark proximal tubules expressing different injury marker combinations. **B** Representative overview and inset images of TAL injury showing NKCC2 (green, encoded by *SLC12A1*), MMP7 (yellow) and DAPI (blue). Arrowheads mark TAL cells expressing MMP7 (TAL Injury h2) in medullary portions of the biopsy. **C** Representative overview and inset images of inflammatory microenvironments around PT Injury h2 cells comprising T-cells (CD8A, magenta), macrophages (CD68, orange) and fibroblasts (ACTA2, green). Injured PT cells are identified by expression of LRP2 (red) and KIM1 (cyan). Nuclei are stained with DAPI (blue). Asterisks mark uninjured proximal tubules. **A**–**C** Scale bars of overview images: 1000 μm. Scale bars of insets: 50 μm. All experiments repeated three times with similar result.

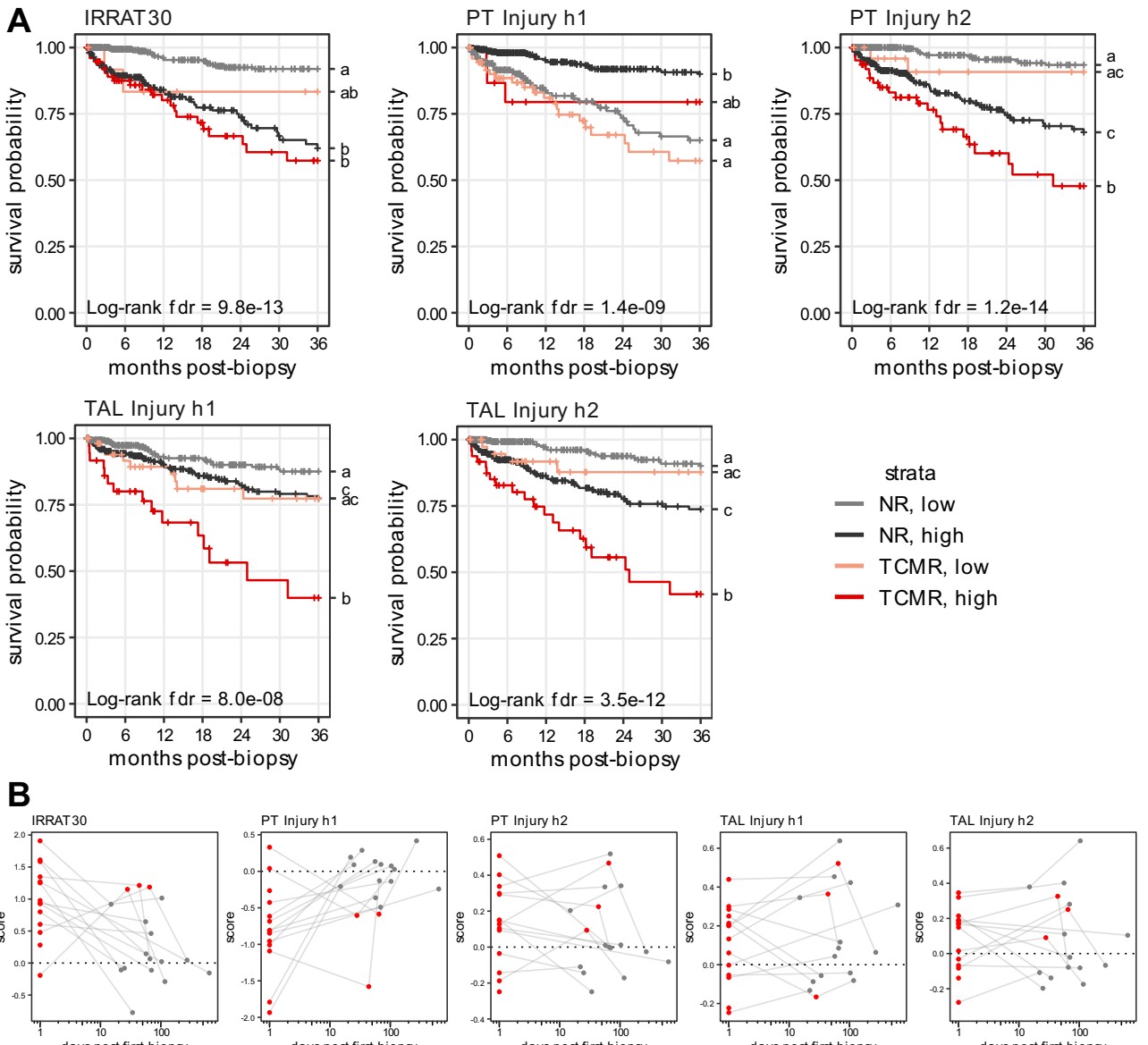

**Fig. 8 | Injured epithelial cell states impact kidney allograft survival after TCMR. A** Kaplan Meier curves for allograft survival in cohort of 1061 kidney transplant biopsies from 1061 patient including 95 TCMR biopsies and 624 no rejection (NR) biopsies. Survival is plotted for TCMR patients below (light red) and above (red) and NR patients below (gray) and above (black) the median score of the respective gene set in the full cohort. False discovery rates (fdr) were derived from log-rank test. Identical letters on the right-hand side of the plot indicate no statistical difference. **B** Gene set scores over time for patients with initial diagnosis of TCMR and TCMR or NR in the follow up biopsies ($n = 12$). Red dots indicate TCMR biopsies and gray dots indicate biopsies without rejection. Source data are provided as a Source Data file.

possibly linked to hemodynamic or metabolic stress. Its slightly lower relative abundance in TCMR could reflect proportional shifts in deconvolution estimates driven by the strong expansion of more severely injured TAL Injury h2 states. We therefore consider TAL Injury h1 a non-TCMR-induced injury phenotype with adverse prognostic implications that warrants further investigation.

Comparing cell states across studies is challenging due to variability in methodologies and limited single-cell studies in transplantation, which typically include fewer patients than studies on native kidneys. Nonetheless, similar cell types have been reported in other contexts including kidney transplantation[39–44]. Whether these represent identical cell states or share superficial similarities with profound underlying differences remains uncertain. However, it is evident that a range of kidney diseases and complications, in, both, native and transplanted kidneys, induces highly dedifferentiated cell states in the

PT and TAL. Taken together, the single-cell deconvolution results (Fig. 6I), the analysis of human AKI samples and the marker gene profiles of the severely injured epithelial cell states in PT and TAL suggest that these states likely represent general maladaptive epithelial programs rather than TCMR-specific ones. While this may limit their diagnostic specificity for TCMR, their prognostic relevance is likely to remain unaffected.

The key question is what these cell states signify. If they merely reflect nephron loss, their correlation with clinical outcomes is unsurprising. However, emerging evidence suggests that injured epithelial cell states may persist and interact with fibroblasts and immune cells, forming pro-fibrotic microenvironments that exacerbate renal damage[25,38]. We demonstrated the presence of such potentially proinflammatory microdomains through unbiased spatial clustering and immunofluorescence validation for PT Injury m4 and its human

counterpart PT Injury h2. We observed that leukocyte proximity to TAL Injury m3 cells (the mouse counterpart to TAL Injury h2) is significantly less frequent than to PT Injury m4 cells (the mouse counterpart to PT Injury h2). This suggests that the pathogenesis of PT and TAL injury may differ.

To explore persistence of injured PT and TAL cell states, we analyzed follow-up biopsies from a subset of patients initially diagnosed with TCMR. In an exploratory effort with a limited sample size, we observed that elevated gene set scores indicative of PT and TAL injury states often persisted, even after the clinical resolution of TCMR. This challenges the notion that EMT-like states, such as PT Injury h2 and TAL Injury h2, simply represent dying tubules. Further research with more patients is needed to confirm these findings and elucidate the conditions under which these injured cell states tend to persist.

Our human TCMR single-cell data, along with the results from bulk transcriptomics on follow-up biopsies, show that PT and TAL injury abundances vary between patients and appear to persist over time in some cases. This suggests that biomarkers indicative of these injured cell states could be used to stratify TCMR patients by risk and to monitor the effectiveness of rejection therapies longitudinally. Such diagnostics could be achieved through analysis of urinary epithelial cells and cost-effectively implemented using fluorescence-activated cell sorting, which we have previously applied successfully in AKI[45]. Furthermore, if these injury states persist, PT and TAL injury cells could serve as therapeutic targets. This, however, warrants careful evaluation in future functional studies beyond the scope of the present work. It also needs to be elucidated if and how additional patient factors such as comorbidities, treatment regimens or other confounders influence the abundances of the investigated cell states.

In summary, our study demonstrates the value of integrating high-resolution molecular data from mouse models with defined injury timelines and analogous data from humans, coupled with large cohorts featuring clinical follow-up data. This approach enables the identification of molecular signals specific to injury populations in single-cell transcriptomics and their correlation with clinical end points. Such strategy helps dissecting the diverse sources of injury in human kidney transplants and opens new avenues for precision medicine.

## Methods

### Animal experiments
Male C57BL/6 and BALB/c mice, aged 10–12 weeks, were used for all the experiments to minimize variability related to hormonal cycles in female mice which can influence the immune responses to the transplant. The animals were sourced from Janvier (Le Genest St Isle, France) and maintained under standard housing conditions at 21–24 °C temperature, 45–65% humidity and 12/12-h dark/light cycle, with *ad libitum* access to food and water. Owing to the exploratory nature of the study, randomization and blinding were not performed. All animal procedures were performed following the Directive 2010/63/EU, the German Tierschutz-Versuchstierverordnung. Ethical approval was obtained from the Regional Ethics Committee for Animal Research (Landesamt für Gesundheit und Soziales Berlin, approval number: G0236/18).

### Human samples snRNA-seq
For human samples, data collection and analysis were performed with informed consent of the patients and with approval of the institutional review board of Hannover Medical School (no 2765). All human study participants provided written informed consent for publication of potentially identifiable medical information presented in supplementary table 1. This included variables such as age and sex. Exclusion criteria for the protocol biopsy program were lack of patient consent, relevant bleeding risks and anticoagulation therapy for an artificial heart valve.

A total of 6 patient samples were included coming from 4 male and 2 female participants (age range 49-77, see Suppl. Table S1). Sex assigned at birth was extracted from clinical records. No gender identity data were collected. Sex-based analyses were not carried out because sample size in each subgroup was insufficient for meaningful stratification.

### Statistical testing
All statistical tests were two-sided unless explicitly stated. The specific test used is indicated in the corresponding figure legends.

### Mouse kidney transplantation
Kidney transplantations were performed in syngeneic (C57BL/6 to C57BL/6 or BALB/c to BALB/c) and allogeneic (C57BL/6 to BALB/c or BALB/c to C57BL/6) mouse combinations under isoflurane inhalation anesthesia as previously described[33]. Briefly, after a midline abdominal incision, the left kidney, aorta and inferior vena cava of the donor mouse were exposed and carefully mobilized. The kidney was flushed in situ with histidine-tryptophane-ketoglutarate (HTK) solution (Custodiol®, Dr. Franz Köhler Chemie GmbH, Bensheim, Germany), then procured *en bloc* with the renal vein, renal artery (with a small aortic cuff) and ureter. The harvested kidney was stored in ice-cold HTK solution for 1 h and thereafter implanted in the left nephrectomized recipient mouse, below the level of native renal vessels. Blood supply to the graft was established through end-to-side anastomoses of the donor renal vessels to the recipient's abdominal aorta and inferior vena cava. Reconstruction of the urinary tract was achieved by directly anastomosing the donor ureter to the recipient's bladder. The contralateral native kidney of the recipient mouse was removed 24 h prior sacrifice on post-operative day (POD) 7 to evaluate renal graft function. For animals undergoing survival analysis, the contralateral native kidney was removed on POD7 and the surviving animals were sacrificed on POD28.

### Assessment of mouse renal graft function
Renal graft function was assessed by serum levels of creatinine and urea. Serum samples were collected terminally from the recipient mice and stored at -20 °C until creatinine and urea were measured using the CREP2 Creatinine Plus version 2 and Urea/BUN assays, respectively, on a Roche/Hitachi Cobas C 701/702 system (Roche Diagnostics, Mannheim, Germany).

### Histopathology
Kidney samples were fixed in 4% neutral-buffered paraformaldehyde for 22–24 h before standard histological processing. Paraffin-embedded tissues were sectioned into 2 µm slices, deparaffinized and stained with periodic acid-Schiff (PAS) stain for histological evaluation. Lesions were evaluated and scored according to the criteria outlined in the Banff classification[46].

### Immunofluorescence staining on mouse kidneys
For immunofluorescence staining (IF), 1.5 µm paraffin-embedded tissue sections were first deparaffinized by sequential incubations in xylene, ethanol and distilled water. Antigen retrieval was performed using 1× Target Retrieval Solution (TRS, Cat. #S1699, Agilent Technologies, Inc., USA) in a pressure cooker (WMF). After antigen retrieval, tissue sections were encircled with a PAP pen, blocked with 5 % skimmed milk or 5 % BSA in 1x TBS-T for 1 h at room temperature.

Primary antibodies (see Table 1 below) diluted in Antibody Diluent (Cat. #S3022, Agilent Technologies, Inc., USA) were applied to the tissue. Incubation was carried out overnight at 4 °C in the dark. After three washes in 1x TBS-T, secondary antibodies (see Table 1 below) and DAPI (1:250 and 1:2500 respectively, diluted in Antibody Diluent) were applied for 1 h at room temperature in the dark. Slides were again washed three times with 1x TBS-T before mounting with 60 % glycerol in PBS supplemented with 584 mM sucrose. Sections were analyzed and photographed with an Eclipse Ti2-A microscope, DS-Qi2 camera and NIS-Elements software (Nikon, USA).

**Table 1 | Antibodies used for immunostaining on mouse kidneys. Individual clones are listed (if applicable) below the antibody names as well as concentration of the stock solution**

| REAGENT or RESOURCE | SOURCE | Dilution | IDENTIFIER |
|---|---|---|---|
| **Primary Antibodies** | | | |
| **Goat polyclonal anti-KIM1** 0,2 mg/mL | R&D Systems | 1:250 | Cat. # AF1817 RRID:AB_2116446 Lot: KCA0319011 |
| **Goat polyclonal anti-NKCC2** 0,5 mg/mL | Abcam | 1:500 | Cat. # ab240542 RRID:AB_2910116 Lot:GR3281474-6 |
| **Mouse monoclonal anti-ANXA3** Clone OTI1A5 1 mg/mL | OriGene | 1:200 | Cat. # TA502123 RRID:AB_11125442 Lot:F001 |
| **Mouse monoclonal anti-LRP2/Megalin** Clone CD7D5 1 mg/mL | Abcam | 1:500 | Cat. # ab184676 RRID:AB_2910117 Lot:GR3194931-13 |
| **Rabbit polyclonal anti-ACSM3** 300 µg/mL | Proteintech | 1:200 | Cat. # 10168-2-AP RRID:AB_2222699 Lot:00010499 |
| **Rabbit polyclonal anti-MMP7** 600 µg/mL | Proteintech | 1:200 | Cat. # 10374-2-AP RRID:AB_2144452 Lot:00101692 |
| **Rabbit monoclonal anti-STK39** Clone OD9L5 0,54 mg/mL | Thermo Fisher Scientific | 1:200 | Cat. # MA5-38038 RRID:AB_2897956 Lot:79478419 |
| **Rabbit monoclonal anti-Vimentin** Clone EPR3776 0,268 mg/mL | Abcam | 1:250 | Cat. # ab92547 RRID:AB_10562134 Lot:GR3258719-12 |
| **Rat monoclonal anti-CD8a** Clone 4SM15 0,5 mg/mL | Thermo Fisher Scientific | 1:250 | Cat. # 13-0808-82 RRID:AB_2572771 Lot:2514270 |
| **Rat monoclonal anti-F4/80** Clone A3-1 1 mg/mL | Bio-Rad | 1:100 | Cat. # MCA497GA RRID:AB_323806 Lot:158948 |
| **Rat polyclonal anti-LRP2/Megalin** 0,25 mg/mL | BiCell Scientific | 1:500 | Cat. # 31012 RRID:AB_3711713 Lot: n/a old aliquot |
| **Rat monoclonal anti-VCAM1/CD106** Clone 429 0,5 mg/mL | Invitrogen | 1:200 | Cat. # 14-1061-82 RRID:AB_467419 Lot:2945393 |
| **Secondary Antibodies** | | | |
| **Donkey anti-goat Alexa Fluor 488** 1,5 mg/mL | Jackson Immuno Research Labs | 1:250 | Cat. # 705-545-147 RRID: AB_2336933 Lot:143223 |
| **Donkey anti-goat Alexa Fluor 594** 1,5 mg/mL | Jackson Immuno Research Labs | 1:250 | Cat. # 705-585-147 RRID: AB_2340433 Lot:142483 |
| **Donkey anti-mouse Cy3** 1,5 mg/mL | Jackson Immuno Research Labs | 1:250 | Cat. # 715-165-150 RRID: AB_2340813 Lot:144189 |
| **Donkey anti-rabbit Alexa Fluor 488** 1,5 mg/mL | Jackson Immuno Research Labs | 1:250 | Cat. # 711-545-152 RRID:AB_2313584 Lot:142192 |
| **Donkey anti-rat Cy3** 1,5 mg/mL | Jackson Immuno Research Labs | 1:250 | Cat. #712-165-150 RRID:AB_2340666 Lot: 143560 |
| **Goat anti-mouse Alexa Fluor 488** 1,5 mg/mL | Jackson Immuno Research Labs | 1:250 | Cat. # 115-545-003 RRID:AB_2338840 Lot: 143649 |
| **Goat anti-rabbit Alexa Fluor 647** 1,5 mg/mL | Jackson Immuno Research Labs | 1:250 | Cat. # 111-607-008 RRID:AB_2632470 Lot: 143347 |
| **Goat anti-rabbit Cy3** 1,5 mg/mL | Jackson Immuno Research Labs | 1:250 | Cat. # 111-165-003 RRID: AB_2338000 Lot: 142739 |

For antibody stripping, sections were washed once in 1x TBS-T, followed by renewed antigen retrieval as described above. Subsequent immunofluorescence staining was then repeated with alternative primary antibodies. Multichannel images were aligned and overlaid using BigStitcher in FIJI/ImageJ. After registration via Load & Stitch Multiple XMLs, fused overlays were exported as TIFFs using Export Transformed Images from BigDataViewer.

**Immunofluorescence staining on human kidneys**

Kidney biopsy sections were obtained from paraffinized biopsy samples stored at the pathology department of Hannover Medical School. TCMR biopsies were obtained due to clinical allograft function decline (indication biopsies). Stable allograft biopsies were obtained as part of the protocol biopsy program 12 months after transplantation. All biopsies were performed between 2014 and 2025. Paraffin-embedded sections were deparaffinized by sequential incubations in ROTI®Histol (Cat. #6640.4, Carl Roth GmbH + Co. KG, Germany), ethanol and distilled water. Heat-induced antigen retrieval was performed by immersion in 9,5 µM citric acid (Cat. #1.00244.0500, Sigma-Aldrich, USA) buffer (pH 6.0) and heating for 2 × 8 min in a microwave. After antigen retrieval, tissue sections were encircled with a PAP pen and blocked with 5 % skimmed milk in 1x TBS-T for 1 h at room temperature. Primary antibodies (see Table 2) were incubated overnight at 4 °C, followed by three washes in 1x TBS-T at room temperature. Secondary antibodies (see Table 2) were incubated for 1 h in the dark at room temperature and subsequently washed three times in 1x TBS-T at room temperature. Primary and secondary antibodies were diluted in 1% BSA in 1x TBS-T. Sections were mounted using Immunoselect Antifading Mounting Medium containing DAPI (Cat. # DNA-SCR-038448, dianova, Germany). Subsequent confocal imaging was performed on a Zeiss LSM980 laser scanning microscope (Carl Zeiss AG, Germany). Antibody stripping was performed as previously described by Gendusa et al.[47]. In brief, sections were washed once in 1x TBS-T followed by incubation in antibody elution buffer containing 2% SDS (Cat. #CN30.2, Carl Roth GmbH + Co. KG, Germany) and 0,8% 2-Mercaptoethanol (Cat. #M3148, Sigma-Aldrich, USA) in 62,5 µM Tris-HCL (Cat. #9090.3, Carl Roth GmbH + Co. KG, Germany) buffer for 30 min at 56 °C. Subsequently, sections were washed in distilled water for 1 h at room temperature with water changes every 15 min. Successful removal of antibodies was confirmed via epifluorescence microscopy. Staining was then repeated with alternative primary antibodies. Multichannel images were aligned and overlaid using BigWarp from the BigDataViewer plugin in FIJI/ImageJ.

**Single nucleus mRNA sequencing mouse and human specimens**

For single-nucleus RNA sequencing (snRNA-seq), the recipient mice were transcardially perfused with ice-cold PBS to clear blood from renal grafts. 1–2 mm middle slices was consistently extracted from the grafts and preserved in pre-cooled RNAlater (Invitrogen #AM7020) at 4 °C for 24 h before being stored at -80 °C until nuclei isolation, following the protocol described by Leiz, Hinze et al.[48].

All samples underwent single-cell sequencing using the 10x Genomics Chromium Next GEM Single Cell 3' v3.1 chemistry protocol (#CG000204 Rev D), targeting 9000–10,000 nuclei per sample. Libraries were sequenced on Illumina HiSeq 4000 platforms (paired-end) and digital expression matrices were generated with the 10x Genomics Cell Ranger software (version 3.0.2) using the parameter '−force-cells 10000' against the mouse mm10 genome.

Archived human kidney biopsy samples were prepared similarly, stored at −80 °C in RNAlater and included from the archive of the Hannover Medical School protocol biopsy program. All subsequent steps, starting from the -80 °C storage in RNAlater, were identical to those used for mouse tissue.

**Table 2 | Antibodies used for immunostaining on human kidneys. Individual clones are listed (if applicable) below the antibody names as well as concentration of the stock solution**

| REAGENT or RESOURCE | SOURCE | DILUTION | IDENTIFIER |
|---|---|---|---|
| **Primary Antibodies** | | | |
| **Goat Polyclonal SLC12A1/NKCC2 antibody** 0,5 mg/ml | Abcam | 1:200 | Cat. # ab240542 RRID: AB_2910116 LOT: 1001309-11 |
| **Goat Polyclonal TIM-1/ KIM-1/HAVCR Antibody** 0,2 mg/ml | R&D Systems | 1:100 | Cat. # AF1750 RRID: AB_2116561 LOT: JTB0924091 |
| **Mouse Monoclonal ANXA3 Antibody** Clone OTI1A5 1 mg/ml | OriGene | 1:100 | Cat. # TA502123 RRID: AB_11125442 LOT: F001 |
| **Mouse Monoclonal CD68 Antibody** Clone KP1 0,41 mg/ml | Dako / Agilent | 1:100 | Cat. # M0814 RRID: AB_2314148 LOT: 00037699 |
| **Mouse Monoclonal VCAM-1 Antibody** Clone 1.4C3 0,2 mg/ml | Thermo Fisher Scientific | 1:25 | Cat. # MA5-11447 RRID: AB_10979792 LOT: AC410113 |
| **Rabbit Monoclonal CD8 alpha Antibody** Clone SI18-01 1 mg/ml | Thermo Fisher Scientific | 1:100 | Cat. # MA5-32069 RRID: AB_2809363 LOT: 79547163 |
| **Rabbit Monoclonal STK39 Antibody** Clone 0D9L5 0,54 mg/ml | Thermo Fisher Scientific | 1:100 | Cat. # MA5-38038 RRID: AB_2897956 LOT: 79475464 |
| **Rabbit Polyclonal ACSM3 Antibody** 0,167 mg/ml | Proteintech | 1:100 | Cat. # 10168-2-AP RRID: AB_2222699 LOT: 00010499 |
| **Rabbit Polyclonal MMP7 Antibody** 0,6 mg/ml | Proteintech | 1:200 | Cat. # 10374-2-AP RRID: AB_2144452 LOT: 00101692 |
| **Rat Polyclonal LRP2 (megalin) Antibody** 0,35 mg/ml | BiCell | 1:100 | Cat. # 31012 RRID: AB_3711713 LOT: 240276 |
| **Anti-alpha smooth muscle Actin antibody** Clone 1A4 0,2 mg/ml | Abcam | 1:100 | Cat. # ab7817 RRID: AB_262054 LOT: not available, completely aliquoted |
| **Secondary Antibodies** | | | |
| **AlexaFluor™ Goat anti-Mouse 488** 2 mg/ml | Thermo Fisher Scientific | 1:500 | Cat. # A11001 RRID: AB_2534069 LOT: 2610355 |
| **AlexaFluor™ Donkey anti-Mouse 555** 2 mg/ml | Thermo Fisher Scientific | 1:500 | Cat. # A31570 RRID: AB_2536180 LOT: 1984063 |
| **AlexaFluor™ Donkey anti-Mouse 647** 2 mg/ml | Thermo Fisher Scientific | 1:500 | Cat. # A31571 RRID: AB_162542 LOT: 819571 |
| **AlexaFluor™ Donkey anti-Rabbit 488** 2 mg/ml | Thermo Fisher Scientific | 1:500 | Cat. # A21206 RRID: AB_2535792 LOT: 2330673 |
| **AlexaFluor™ Donkey anti-Rabbit 555** 2 mg/ml | Thermo Fisher Scientific | 1:500 | Cat. # A31572 RRID: AB_162543 LOT: 2088692 |
| **AlexaFluor™ Goat anti-Rabbit 647** 2 mg/ml | Thermo Fisher Scientific | 1:500 | Cat. # A21245 RRID: AB_2535813 LOT: 2497486 |
| **AlexaFluor™ Donkey anti-Goat 488** 2 mg/ml | Thermo Fisher Scientific | 1:500 | Cat. # A11055 RRID: AB_2534102 LOT: 1737907 |
| **AlexaFluor™ Donkey anti-Goat 555** 2 mg/ml | Thermo Fisher Scientific | 1:500 | Cat. # A21432 RRID: AB_2535853 LOT: 1249013 |
| **AlexaFluor™ Donkey anti-Rat 488** 2 mg/ml | Thermo Fisher Scientific | 1:500 | Cat. # A21208 RRID: AB_2535794 LOT: 1744717 |
| **AlexaFluor™ Goat anti-Rat 555** 2 mg/ml | Thermo Fisher Scientific | 1:500 | Cat. # A21434 RRID: AB_2535855 LOT: 1304741 |

## Xenium in Situ Profiling

In situ single cell RNA expression analysis was performed using the Xenium system (10X Genomics). 5μm thick FFPE tissue sections were placed on a Xenium slide according to the Xenium Tissue Preparation Guide (CG000578). Sections were dried at 37 °C for 2 h and placed overnight in a desiccator at room temperature, followed by deparaffinization and decrosslinking (CG000580). The Probe Hybridization Mix was prepared using a pre-designed 379 gene panel (Mouse Tissue Atlassing v1) and 100 gene custom add-on panel (Suppl. Data 1). The 100 custom genes were selected based on previous mouse single cell sequencing experiments and markers published in the literature. They include canonical marker genes for all expected kidney major cell types as well as expected cell populations induced by kidney injury. Probe Hybridization, Ligation & Amplification were performed according to the Xenium In Situ Gene Expression user guide (CG000582). Raw data was processed in real-time with on-board Xenium Analyzer software v1.7.6.

## Bioinformatic analyses of spatial transcriptomic data

**Determination of major cell types in xenium data.** Raw data was analyzed using the xenium ranger v2.0 import-segmentation tool with parameter –expansion-distance=5. Results from this were directly imported into Seurat.

Cell types were annotated by label transfer from snRNA-seq using SingleR v2.0.0. We first assigned major cell types with the function SingleR(test = as.SingleCellExperiment(spatial data), ref = snRNA-seq data, labels = cell type annotations from snRNA-seq, aggr.ref = TRUE, de.method = "wilcox") and retained only labels within pruned.labels. The same procedure was subsequently applied within each major cell type to refine the annotations.

**Spatial proximity.** For every injury cluster of interest (e.g., PT Injury h2), the co-occurence to each leukocyte (sub) cell type was calculated with squidpy (v1.4.1) using Python (v3.10.14)[49]. Briefly, squidpy.gr.co_occurrence was run with a distance interval of 5 to 50 μm with step size 5 (and a second longer distance from 25 to 250 μm with stepsize 25) based on the provided centroid information from the Xenium data. We chose 25 μm as indicative of being a direct neighbor to calculate a one sample t-test for the significant deviation of the co-occurrence from a random co-occurence (ratio 1) across the allogeneic samples. Multiple testing correction was performed with the Benjamini-Hochberg procedure.

**Spatial domains.** To identify spatial domains in an unsupervised manner SpatialLeiden (v0.2.0) was used[36]. First, a directed spatial nearest neighbor graph was built per sample using squidpy.gr.spatial_neighbors (v1.4.1) with coord_type 'generic' and 10 neighbors. The distance between cells was transformed to a connectivity using spatialleiden.distance2connectivity. The undirected gene expression nearest neighbor graph was generated across all samples with scanpy.pp.neighbors (v1.10.1)[50] using the Harmony batch-corrected latent space. Domains were identified using spatialleiden with a resolution of 0.9 and a layer_ratio of 1.1 running for 10 iterations.

To gain insights in the localization of TAL subtypes, the medullary domain (domain 2) was subclustered with spatialleiden using a resolution for the latent space neighbor graph and the spatial neighbor graph of 0.4 and 0.8, respectively. Similarly, the cortical domains containing the majority of the PT cells and exhibiting similar composition but varying levels of immune infiltration (domain 0, 1 and 6) were subclustered with spatialleiden using resolutions of 0.5 and 0.7, respectively.

## Bioinformatic analyses of single nucleus mRNA sequencing data

**Cell type annotation and initial clustering.** Digital expression matrices were generated with the 10x Genomics Cell Ranger software (version 3.0.2 for mice and version 8.0.0 for human samples) using the

parameter '--expect-cells 10000' against the mouse mm10 or human GRCh38-3.0.0 genome. Human samples were further filtered using cellbender remove-background with fpr 0.01 to remove ambient RNA.

Data integration and clustering were performed using the following workflow in Seurat version 5.1.0: NormalizeData -> FindVariableFeatures -> SelectIntegrationFeatures -> FindIntegrationAnchors -> IntegrateData -> ScaleData -> RunPCA(npcs=30) -> harmony::RunHarmony(group.by.vars = c("group"), lambda = 2, tau = 1000, theta = 1, assay.use = "integrated", kmeans_init_nstart = 60, kmeans_init_iter_max=2000, max.iter.harmony = 30) -> ScaleData -> FindNeighbors(reduction = "harmony", dims=1:15) -> FindClusters(resolution=0.5).

Cell types were assigned based on marker gene expression as published previously[22]. For the PT, injury population PT Injury m1 was finally determined by increasing clustering resolution to 0.6.

**Differential gene expression analysis.** For mouse samples, differential gene expression analysis was performed using Seurat's FindMarkers function, comparing allogeneic kidneys from each group (C57BL/6 and BALB/c) to their syngeneic controls per cell type. Genes were considered differentially expressed in case of $|\log 2$ fold change $| > 1$, adjusted $p$ value $< 0.05$ and expressed in at least 10% of cells in allogeneic kidneys (for differentially upregulated genes in allogeneic kidneys) or 10% of cells in syngeneic kidneys (for downregulated genes in allogeneic kidneys).

**Pathway analysis of DGE genes using WGCNA modules.** For both strain comparisons (C57BL/6 to BALB/c vs. C57BL/6 to C57BL/6 and vice versa), pathway analysis was performed on the same sets of significantly upregulated genes described in the main DGE analysis (Fig. 2A; log2FC > 1, adj. $p < 0.05$). To group DGE genes into functionally coherent clusters, we applied *Weighted Gene Co-expression Network Analysis* (WGCNA; blockwiseModules, networkType = "signed", minModuleSize = 20, deepSplit = 3, mergeCutHeight = 0.20). Genes were ordered by module and displayed in clustered heatmaps (cell types × genes; log2FC clipped to [0, 3] for color scaling). Each module was tested for pathway enrichment using *clusterProfiler*, with MSigDB Hallmark gene sets (Mus musculus) for a stringent run (q < 0.05) and Hallmark + Reactome for a broader overview (q < 0.1). Results were summarized as module×pathway dot plots to support interpretation of the DGE gene sets at the pathway level.

**Trajectory inference with PAGA.** For PT and TAL mouse subclusterings, trajectory inference was performed using partition-based graph abstraction (PAGA) in Scanpy version 1.11.4. For each dataset, selected healthy subclusters were grouped into a single starting population (healthyPT or healthyTAL) in the largersubct metadata field. Log-normalized expression values were used to identify the top 500 highly variable genes (flavor = "seurat"), followed by PCA, neighborhood graph construction and diffusion map embedding. PAGA graphs were computed on the largersubct groups and diffusion pseudotime was calculated with the root set to the "healthy" cluster and normalized to a 0–1 range for visualization. The resulting PAGA connectivity graphs were visualized with nodes colored by pseudotime.

**Cross-species analysis.** Human-to-mouse mapping was performed for PT and TAL subclusters using SingleR v2.0.0, with human single-cell datasets serving as reference. Mouse genes were converted to unique human orthologs via biomaRt (Ensembl, Dec 2021) and only intersecting orthologous genes were retained. Log-normalized expression values were extracted from both species and human annotations were used as reference labels. SingleR was run with default settings and pruning enabled to remove ambiguous matches. Mapping performance was evaluated by confusion matrices comparing original mouse subclusters with SingleR-derived human-based labels, followed by column-wise normalization and visualization as non-clustered heatmaps.

**Determination of marker genes for injury clusters.** To get marker genes specific for the respective cell populations (further referred to as TP = "target population"), we first generated a list of potential marker genes for the TP by using the FindMarkers function with ident.1 = TP and ident.2 = "all other cells". We then calculated the average expression of genes of interest in the TP. We also calculated the average expression of the mentioned genes in the adjacent injured cell population. E.g., if TP was TAL Injury h2, we also calculated average gene expression in TAL Injury h1. We only considered genes which show higher average expression in TP than in the adjacent injury population. This reduces the number of genes and the computational resources required downstream.

The last filtering step included the generation of 100-cell neighborhoods of nearest neighbors around each cell not in TP. We furthermore excluded cells with 100-cell neighborhoods overlapping with the TP. We then calculated average expression of each of the remaining gene candidates in all 100-cell neighborhoods. Final genes only included genes which showed an average expression in the TP > 1.25 fold compared to all other 100-cell neighborhoods.

**Kidney transplant bulk transcriptomics cohort.** We included a total of 5086 kidney transplant biopsies. 3570 biopsies from 3995 patients were collected through International Collaborative Microarray Extension Study (INTERCOMEX; NCT01299168) and Trifecta-Kidney (NCT04239703) clinical trials and MMDx-Kidney sub-studies. An additional 1516 biopsies, included from MMDx service laboratory in Portland, OR (Kashi Laboratories, submitted as anonymized files for this study), were processed for MMDx. Genome-wide assessment (49,495 probe set values representing 19,462 genes) of the 5086 biopsies was measured by microarrays[51]. MMDx signouts of NR, TCMR, ABMR and Mixed were available for 4371 biopsies. Rejection and injury states were derived according to previously published methods[52]. Biopsies having ≤10% cortex were excluded leaving 3858 biopsies. From this set of 3858 biopsies, MMDx signouts and graft status were available for 1292 biopsies from 1061 patients. For survival analyses, we selected the first biopsy post-transplant for each patient, leaving 1061 biopsies for assessment. For deconvolution analyses, cell composition was estimated in the cohort of 3858 biopsies. The MMDx-Kidney studies adhere to the Declaration of Helsinki. All biopsies were collected with informed consent per institutional review board review at each local center and approved in Edmonton by the University of Alberta (#Pro00022226). The clinical and research activities being reported are consistent with the Principles of the Declaration of Istanbul as outlined in the 'Declaration of Istanbul on Organ Trafficking and Transplant Tourism'.

**Conservation in gene expression profiles of TCMR in single nuclei and bulk transcriptomic data.** Agreement between gene expression profiles for TCMR among mouse blk6 single nuclei, human single nuclei and human bulk transcriptomics assays was assessed using Spearman correlation coefficients in log2FC values for the top 500 increased and decreased genes in allogenic vs syngeneic (mouse single nuclei) TCMR vs stable (human single nuclei) and TCMR vs NR (human bulk).

**Conservation in gene expression profiles of indication and protocol biopsy bulk transcriptomic data.** Agreement between gene expression profiles for TCMR among human indication and protocol biopsies assessed by bulk transcriptomics was assessed using Spearman correlation coefficients in log2FC values for the top 500 increased and decreased genes in TCMR-indication vs NR and TCMR-protocol vs NR (N TCMR protocol = 22, N TCMR indication = 140).

**Gene set scores in bulk transcriptomic data.** Gene set scores for each human PT and TAL cell state were assigned to each biopsy in the microarray population by taking the geometric mean expression of all marker genes for each biopsy, normalized to the geometric mean

expression in 4 nephrectomy control biopsies (i.e., mean score in nephrectomy is 0). Risk of graft loss 3-years post biopsy associated with gene set scores was assessed for NR, TCMR, Mixed and ABMR populations, including a total of 1061 patients from 605 male, 353 female and 103 unreported sex participants (age range 2-92 years). Sex assigned at birth was voluntarily provided by each patient's clinician. No gender identity data were collected. Scores were assessed individually with Kaplan-Meier estimates, stratified by their median value in the full population. Sex had no significant impact on survival in this cohort and was omitted from the final Kaplan-Meier analyses to reduce over stratification and increase the number of failures in each stratum.

The transition of gene set scores following TCMR diagnosis was depicted in MMDx kidney biopsies in patients diagnosed with TCMR (n = 12) which eventually progressed to no rejection (NR) in subsequent follow-up biopsies.

**Cellular deconvolution in bulk transcriptomic data.** Cell compositions were estimated using the CIBERSORTx platform[53–55]. A de novo signature matrix was produced using the gene expression profiles of the 28 cell types identified in the human single nuclei studies described above. The cibersortx/fractions module was used to produce the signature matrix with the following parameters: single_cell = TRUE, g_min = 300, g_max = 5000, q_value = 0.05, filter = FALSE, k_max = 999, replicates = 5, sampling = 0.5, fraction = 0.75 with all other parameters set to default. In cases where genes in the signature matrix were represented by multiple Affymetrix probesets, the mean expression of all probsets for each gene was used. CIBERSORTx was run using the cibersortx/fraction module with the following parameters; single_cell = TRUE, perm = 1000, QN = FALSE, rmbatchSmode = TRUE, with all other parameters set to default. Deconvolution was run using S-mode batch correction to account for cross-platform technical variability. A total of 3858 biopsies from 3210 patient samples were included coming from 1019 male, 600 female and 1591 unreported sex participants (age range 2-92 years). Sex assigned at birth was voluntarily provided by each patient's clinician. No gender identity data were collected. Sex-based analyses were not carried out due to data incompleteness in reported sex. All samples were fit appropriately (i.e., $p < 0.05$ from 1000 permutation), amounting to a total of 2155 NR, 298 TCMR, 298 Mixed and 1107 ABMR biopsies available for assessment. Estimates of PT and TAL cells were compared across MMDx groups using arcsin-transformed Bayes-moderated t-tests[56]. Healthy PT and TAL cells were pooled prior to group testing.

### Ethics statement

All research in this study complies with legal and ethical regulations governing both animal experiments (Landesamt für Gesundheit und Soziales, Berlin, Germany) and the collection and processing of human specimens (snRNA-seq: Hannover Medical School, Hannover, Germany and bulk transcriptomics: University of Alberta, Edmonton, Canada). The corresponding approvals and approving institutions are listed in the respective sections below. Patient participants received no compensation; individual patient approval was necessary for inclusion in this study and included publishing individual age and sex.

### Reporting summary

Further information on research design is available in the Nature Portfolio Reporting Summary linked to this article.

## Data availability

The snRNA-seq and ST data generated in this study have been deposited in the Gene Expression Omnibus database under accession code GSE284742. Source data are provided with this paper. Note that for some graphs derived from snRNA-seq and spatial transcriptomics data, the underlying source data are not included in the Source Data files due to their volume. These data are available through the referenced GEO dataset. Source data are provided with this paper.

## Code availability

Code is released online[57] under: https://doi.org/10.5281/zenodo.17673467.

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

## Acknowledgements

This study was supported by grants to M.I.A., I.M.S., F.A., K.M.S.O.: Deutsche Forschungsgemeinschaft (DFG, German Research Foundation): Project ID 394046635, SFB 1365, grants to C.H.: DFG (grant HI 2238/2–1) and Ministry for Science and Culture of Lower Saxony as project of the Center for Organ Regeneration and Replacement (CORE), Transplant Center, Hannover Medical School and DFG grant to M.F. (FA 845/7-1). L.M.S.G. was funded by a clinician scientist grant from the German Society of Internal Medicine (Deutsche Gesellschaft für Innere Medizin – DGIM). J.R. was supported by the PRACTIS Clinician Scientist Program, funded by Hannover Medical School and the DFG (DFG ME 3696/3).

## Author contributions

A.M.P. performed animal experiments involving mouse kidney transplantation and follow-up analyses, including graft function, histology, graft and animal survival, performed snRNA-seq experiments and provided inputs during manuscript writing. L.J. performed majority of final downstream bioinformatics analyses including snRNA-seq of mouse and human as well as of spatial transcriptomics data, wrote the manuscript

and provided critical input for the remaining analyses. P.T.G. performed single-cell deconvolution analyses and all other analyses with respect to human bulk transcriptomic data and provided critical input for remaining analyses and critical input for manuscript writing. V.A.K. performed mouse immunofluorescence and all other mouse histology stainings and provided critical input for manuscript writing. J.R. performed human immunofluorescence stainings and provided critical input for manuscript writing. N.M.B. performed co-occurrence analysis and spatial domain clustering and provided critical input for manuscript writing. L.M.S.G. performed leukocyte annotation and clustering of mouse and human snRNA-seq data and provided critical input for manuscript writing.JL performed mouse snRNA-seq experiments as well as initial analyses of mouse snRNA-seq data. S.S. helped in animal experiments involving mouse kidney transplantation and follow-up analyses, including graft function and graft and animal survival. I.P. performed Xenium experiments. R.G. supervised human immunofluorescence stainings and sample selection. S.L. provided critical input for study design. J.G. helped in annotation of mouse and human leukocyte subclusters. F.L. helped in annotation of mouse and human leukocyte subclusters. J.S. provided and analyzed histopathology of all human snRNA-seq samples as well as samples for immunofluorescence. J.H.B. provided and analyzed histopathology of all human snRNA-seq samples as well as samples for immunofluorescence. I.S. performed samples selection and clinical characterization of human snRNA-seq samples. I.M.S. conceptualized and co-supervised mouse kidney transplantation experiments and follow-up analyses. F.A. conceptualized and co-supervised mouse kidney transplantation experiments and follow-up analyses. J.A. performed Xenium experiments. T.C. performed Xenium experiments.WG performed samples selection and clinical characterization of human snRNA-seq samples. N.I. performed co-occurrence analysis and spatial domain clustering and provided critical input for manuscript writing. M.F. supervised mouse immunofluorescence and conventional staining and contributed to experimental design and data interpretation. K.M.S.O. conceptualized and co-supervised initial parts of the study (snRNA-seq of mouse samples and initial analyses) and provided critical input during manuscript writing.PFH conceptualized the whole study, provided critical input for paper writing, and supervised the analysis of human bulk transcriptomic samples. M.I.A. conceptualized the whole study, provided critical input for paper writing, supervised and performed mouse kidney transplantation experiments, mouse snRNA-seq and Xenium experiments. C.H. conceptualized the whole study, wrote the paper, performed and/or supervised all final analyses on mouse and human (transcriptomic) data and performed human snRNA-seq experiments.

## Funding

## Competing interests

The authors declare no competing interests.

## Additional information

[1]Department of Surgery, Experimental Surgery, Charité – Universitätsmedizin Berlin, corporate member of Freie Universität Berlin, Humboldt-Universität zu Berlin and Berlin Institute of Health, Berlin, Germany. [2]Department of Nephrology and Hypertension, Hannover Medical School, Hannover, Germany. [3]Alberta Transplant Applied Genomics Centre, Edmonton, AB, Canada. [4]Institute of Translational Physiology, Charité – Universitätsmedizin Berlin, corporate member of Freie Universität Berlin and Humboldt-Universität zu Berlin, Berlin, Germany. [5]PRACTIS Clinician Scientist Program, Dean's Office for Academic Career Development, Hannover Medical School, Hannover, Germany. [6]Berlin Institute of Health at Charité – Universitätsmedizin Berlin, Center of Digital Health, Berlin, Germany. [7]Freie Universität Berlin, Department of Mathematics and Computer Science, Berlin, Germany. [8]Department of Medicine V, University Medical Centre Mannheim, University of Heidelberg, Mannheim, Germany. [9]Genomics Technology Platform, Max Delbrück Center for Molecular Medicine, Berlin, Germany. [10]Core Unit Genomics, Berlin Institute of Health at Charité, Berlin, Germany. [11]Nephropathology Unit, Institute of Pathology, Hannover Medical School, Hannover, Germany. [12]Department of Surgery, Krankenhaus der Barmherzigen Brüder, Graz, Austria. [13]Department of Medicine, Division of Nephrology and Transplant Immunology, University of Alberta, Edmonton, AB, Canada. [14]These authors contributed equally: Anna Maria Pfefferkorn, Lorenz Jahn, Patrick T. Gauthier. [15]These authors jointly supervised this work: Philip F. Halloran, Muhammad Imtiaz Ashraf, Christian Hinze. ✉e-mail: phallora@ualberta.ca; muhammad-imtiaz.ashraf@charite.de; hinze.christian@mh-hannover.de

