## [Transparent Peer Review file · Nature Communications]

Injured epithelial cell states critically impact kidney allograft survival after T-cell-mediated rejection

Corresponding Author: Professor Christian Hinze

Version 1:

Reviewer comments:

Reviewer #1

(Remarks to the Author)

The manuscript focusses on the evaluation of TCMR argument that the molecular pathways triggered by TCMR remain poorly understood. The authors state that although when TCMR is treatable, it associates with reduced allograft survival, suggesting current therapies are not adequately addressing all the aspects of TCMR-induced injury. This statement represents the scientific premise of the study. They evaluate using single cell approaches (snRNAseq and spatial transcriptomics) kidney tissues from two mouse models of TCMR (including allo- and iso-genic kidney transplant mouse models). Although the authors claim that the study aim is to delineate the molecular changes associated with TCMR and evaluate their impact on allograft outcomes, they mainly focused on the analyses of the epithelium response to TCMR-associated injury. Nevertheless, this is the more novel aspect of the study. They compared the findings from mouse data to molecular data from human biopsies. Finally, they use bulk transcriptomics from a large cohort of patients to assess potential clinical relevance for elucidating potential therapeutic targets. The study includes allo- and iso-genic kidney transplant mouse models. Overall, the manuscript is well presented, organized, and test an important hypothesis. The use of single cell approaches, interspecies comparison, and validation using bulk transcriptomics is a strength of the study. Focus on the response of the epithelium to the injury is also relevant. The study effectively links mouse and human datasets, demonstrating conserved injury states in PT and TAL cells. However, some points that deserve further evaluation were identified including:

1- Most of the single cell transcriptomics evaluations seems superficially evaluated. Although cell identity was established for PT subclusters (PT injurym1-4 and PT proliferating), the results about specifics of these clusters are limited. There are no specific results defining these cells subclusters and/or analyses about characterization of cell type-specific pathways. The description of the PT injury m4 cell cluster (identified as most injured cluster by number of injury-associated marker genes), is limited to "The marker genes for PT Injury m4 include several associated with maladaptive repair cells in AKI, such as Vcam1, Cd44 and Vim22,24". Overall, most the results rely on markers previously described in the literature for native kidneys with AKI. Likewise, the same limitations affect the evaluation of TAL cell subclusters.

2- Although the main interest seems to focus on PT and TAL cell clusters, immune cells and epithelium-immune interactions are critical as part of resolution of injury or progression to impaired repair during TCMR. However, the cell type-specific characterization of immune cells is minimal. Specifically, what type of macrophages were involved in injured PT-immune cell interactions? Fig 4 A shows a subcluster analyses of immune cells with 4 macrophages populations. However, no characterization of the macrophage subpopulations is provided nor their associated profiles. Same point is valid for all the immune cells.

3- In the cross-species evaluation, the study effectively bridges mouse and human datasets, demonstrating conserved injury states in PT and TAL cells. However, although injured PT and TAL cells were identified, their relative abundance was not significantly higher in TCMR samples compared to stable allografts. How alike were the PT injured specific profiles between conditions? This point deserves further evaluation/discussion.

4- The human biopsies were obtained from protocol biopsy program at Hannover Medical School, meaning they were not necessarily taken at the time of acute rejection symptoms but rather as part of scheduled monitoring of kidney transplant patients. This contrasts with the controlled timing in mouse models, reducing direct comparability. This is likely reflected by the absence of the cluster of proliferating cells in the human samples. Further discussion about how this issue may affect other cell clusters and moreover, their specific transcriptome must be discussed. Also, this reviewer was not able to identify

the Banff scoring for human samples with TCMR used in the snRNAseq study or the specific time of biopsy collection.

5- The study focused primarily on PT and TAL injury states, but other cell types (e.g., immune cells, endothelial cells, fibroblasts) may contribute significantly to TCMR pathogenesis. Additional cell type-specific analyses could enhance the understanding of injury mechanisms in response to TCMR.

6- The longitudinal analysis included only 12 patients, which is a small sample size to draw definitive conclusions about persistence of injury states post-TCMR.

7- The significant variability in injury-related gene set scores among TCMR patients indicates heterogeneous injury responses, making it challenging to define a universal injury signature. This suggests that additional patient-specific factors (e.g., immune responses, comorbidities, treatment regimens) may influence epithelial injury progression and should be further investigated.

8- Although TAL injury signatures were more pronounced in TCMR, they were not exclusive to it, as they also correlated with injury in AKI. This suggests that while TAL injury is relevant for TCMR outcomes, other forms of allograft stress (e.g., ischemia) may also contribute to its expression.

(Remarks on code availability)

Reviewer #2

(Remarks to the Author)

T-cell-mediated rejection (TCMR) remains a significant hinderance to long-term kidney transplant survival, as it contributes to graft injury and eventual failure despite immunosuppression strategies. To better understand molecular mechanisms associated with TCMR the authors induced acute TCMR in mouse kidney transplant models and analyzed molecular changes using snRNA sequencing and spatial transcriptomics. These findings were compared with human biopsy data from rejected and stable allografts. The study revealed that TCMR causes significant gene expression changes and epithelial injury, particularly in proximal tubules (PT) and thick ascending limbs (TAL), with spatial heterogeneity and proximity to immune cells. Cross-species analysis confirmed similar injury patterns in human TCMR cases. Furthermore, kidney transplant outcomes were linked to the persistence of injured epithelial states, even after rejection resolution. The manuscript reports interesting data with potential value in expanding current understanding about TCMR and underlying molecular landscapes. The results are significant despite the small sample size in the validation cohort. There are a few aspects the authors could do to improve the quality of the manuscript's claims and conclusions and make it more useful to its readers.

The primary concern about the manuscript is that the conclusion of this study is inconclusive in terms of the utility of such gene set scores in enabling clinicians to triage transplant recipients into different bins of TCMR. The sample size is small, and despite the molecular score based on the snRNA seq data from mice and humans, other injury types in transplant have not been discussed at all. The authors have brought AKI and CKD studies, but have failed to discuss how to interpret the data in the context of ABMR and other transplant injuries such as mixed rejection.

Minor:

For Fig 1D, top panel, please provide color codes for major cell types.

Fig 6A-- How would other phenotypes behave other than TCMR and nonTCMR in Kaplan Meier curves? Did nonTCMR cohort include ABMR? How did the gene set score perform for mixed rejection cases?

Fig 6B-- There is no mentioning of the significance of orange vs grey circles in the figure legend or the result section where the data is presented. Please provide the information both in the legend as well as the result section.

(Remarks on code availability)

Reviewer #3

(Remarks to the Author)

In this manuscript, the authors employ a murine kidney transplantation model to investigate epithelial cell injury states during acute T cell-mediated rejection (TCMR). They performed snRNA-seq on kidneys from both syngeneic and allogeneic transplant recipients (BALB/c and C57BL/6), identifying distinct injury-associated subpopulations within proximal tubule (PT) and thick ascending limb (TAL) cells. Spatial transcriptomics (Xenium) was used to characterize immune cell subtypes within the rejecting allografts. In addition, the authors derived injury gene expression signatures from snRNA-seq analysis of human kidney transplant biopsies and correlated these injury-associated signatures with reduced graft survival in an independent bulk transcriptomic dataset.

The core hypothesis—that injured epithelial cell states contribute to allograft dysfunction—is conceptually important, and the integration of scRNA-seq with spatial technologies is a valuable strategy. However, this manuscript fails to meet the analytical and experimental standards necessary to support such claims. The analysis lacks rigor, fails to provide cell type-specific insights, and omits essential validation steps. Several figures are descriptive but not interpretable in their current form. I do not believe the manuscript is suitable for publication. Substantial re-analysis, validation, and clarification are needed to bring the study to the level expected by the field. Below are my specific comments.

1. In figure 2A, if the BALB/c to C57BL/6 kidneys has milder rejection, why there are more DEGs in PC, EC, intC, PEC and

IC-A when comparing to the C57BL/6 to BALB/c kidneys?

2. The heatmap shown in Figure 2B provides minimal information beyond the number of DEGs in each cell type. The authors should highlight key cell type-specific genes, particularly in PT and TAL cells, to improve interpretability. In addition, experimental validation of select disease-associated genes in these cell types would considerably strengthen the study's conclusions.
3. Figure 2C basically shows that all these cell types share most of the disease pathways. Therefore, the authors should conclude that few cell type-specific pathways were identified, which makes the overall conclusion less compelling. The authors should focus more on highlighting any truly cell type-specific disease pathways to strengthen the impact of their findings.
4. In Figure 3A, PT injury m2–3 cannot be classified as distinct injured states, as they share all marker gene expression and lack unique disease genes. If the authors want to define these as transitional injury states, pseudotime trajectory analysis would be more appropriate than clustering analysis.
5. The accuracy of mapping scRNA-seq to Xenium data depends heavily on the gene panel used in Xenium—for example, whether markers for PT subtypes are included. As a result, it is difficult to assess the accuracy of the spatial cell mapping shown in Figure 3A. The authors should apply the same mapping algorithm to PT S1–S3 subtypes to determine whether PT S3 localizes to the corticomedullary junction and PT S1/2 to the outer cortex, as expected. This can serve as internal validation of the approach.
6. In Figure 3B, if PT injury m1 represents an injured cell type, it is unclear how to explain the lack of change in its proportion between allogeneic and syngeneic kidneys. The only marker used to define PT injury m1 is Nqo1. Since this is a novel marker in this context, the authors should provide additional information about the gene, including its primary function and potential role in tubular injury. Furthermore, immunofluorescent staining is needed to confirm the presence of this injury state in the allogeneic kidney.
7. All of the concerns raised above regarding the PT subtypes (points #4–#6) also apply to the TAL subtypes shown in Figure 3C and 3D.
8. Since the authors have generated Xenium data, they should perform subclustering analysis on the PT and TAL populations within the Xenium dataset to validate the injured cell states and assess their spatial distribution in allogeneic versus syngeneic kidneys.
9. It is unclear why the authors focus exclusively on immune cells in Figure 4 for the Xenium data analysis. A more comprehensive analysis should include all cell types to determine whether the same populations identified by scRNA-seq can also be resolved using Xenium data alone. In Supplementary Figure S4A, the authors perform clustering on the Xenium dataset, but substantially fewer cell types are shown, and no subclustering analysis was conducted for PT and TAL.
10. The authors need to provide details on how the 100 custom genes were selected for their Xenium gene panel design. For instance, if the panel lacks sufficient markers for injured PT or TAL subtypes, the entire study would be compromised. In such a case, the key findings from the scRNA-seq data regarding injured PT and TAL cells would be difficult to validate or reproduce using the Xenium platform, making integration between the two datasets unreliable. Furthermore, the absence of appropriate markers would preclude meaningful downstream analyses, such as determining the spatial roles of injured PT and TAL subpopulations. For example, niche analysis—which could reveal which injured states preferentially recruit immune cells—would not be possible, thereby limiting mechanistic insights into tubular injury during TCMR.
11. It is unclear how the authors generated Figure 4D, as they did not perform subclustering of PT and TAL cells. It appears that computational methods were used to map snRNA-defined cell types onto the spatial data. However, this mapping relies on the set of genes shared between the snRNA-seq and Xenium platforms. If the gene panel design lacks sufficient markers for injured states, particularly for PT and TAL, the resulting cell mapping may be highly inaccurate.
12. In addition, for the neighborhood analysis in Figure 4D, the authors simply report the percentage of neighboring PT and TAL cells, which is a descriptive rather than statistical approach. The authors should utilize publicly available tools designed for spatial neighborhood analysis to reanalyze the data more rigorously. To support their findings, they should also provide immunostaining data to confirm that the identified cell types are indeed in close spatial proximity.
13. In Figure 5, the authors focus solely on demonstrating the similarity of cell types between human and mouse in TCMR. However, they fail to present more critical data—specifically, whether the gene expression changes observed between stable allograft and TCMR biopsies in humans are similar to, or distinct from, those observed between allogeneic and syngeneic kidneys in the mouse model. This comparison is essential to establish the translational relevance of the mouse findings to human disease.
14. Since the authors selected different sets of genes in Figure 5C and 5F compared to those used in Figure 3A and 3C, it is unclear whether the human injured cell states truly correspond to the mouse injured cell states. The simple Pearson correlation analysis presented in Figure 5F and 5H is insufficient to support the conclusion that these are the same cell types. Additional analyses are needed to strengthen this claim. For example, the authors should identify injury markers shared between human and mouse tubular subtypes and perform immunostaining to validate the identity of these matched cell states.
15. The authors should use the gene signatures derived from the human biopsy scRNA-seq data to perform cell type deconvolution on the bulk transcriptomic dataset presented in Figure 6. This analysis would allow them to determine whether the proportions of injured tubular subtypes are specifically increased in TCMR or mixed rejection biopsies, but not in ABMR biopsies. Such evidence would provide stronger support for the relevance of the identified injury signatures in distinguishing rejection types.

(Remarks on code availability)

Version 2:

Reviewer comments:

Reviewer #1

(Remarks to the Author)

The authors made a great effort in revising the manuscript and added important new data, but several key issues remain:

- 1-The specificity of the epithelial injury states is overstated, as the so-called "TCMR-associated" states are also enriched in ABMR, mixed rejection, AKI, and CKD. These should be reframed as general maladaptive epithelial programs with prognostic rather than diagnostic value.
- 2-The issue of causality is not established, since the study relies on transcriptional correlations and spatial proximity analyses. Claims of therapeutic targetability without functional perturbation or mechanistic studies this remains speculative and therefore s should be tempered.
- 3- The limitations of bulk transcriptomic analyses remain, because prognostic associations are presented without multivariable adjustment or controls for cell composition, which undermines the independence of the signal.
- 4- The human snRNA-seq dataset is underpowered, with only three TCMR and three stable biopsies. Heterogeneity within PT-h2 (e.g., KIM1/VCAM1 vs. STK39/ACSM3 expression) remains unresolved.
- 5- The ambiguity of the TAL-h1 state persists, as it is prognostically relevant but paradoxically depleted in TCMR and lacks a clear mouse counterpart. Its biological meaning requires clarification.
- 6- The claims about persistence of injury states are insufficiently supported, as they are based on only 12 patients with follow-up biopsies. These findings should be described as exploratory rather than definitive.

(Remarks on code availability)

Reviewer #2

(Remarks to the Author)

The authors have adequately responded to my questions and I do not have further concerns or questions about the manuscript.

(Remarks on code availability)

Reviewer #3

(Remarks to the Author)

The authors have provided substantial data to address my previous concerns in this revised version. The manuscript has been significantly improved, and I have no further comments.

(Remarks on code availability)

Version 3:

Reviewer comments:

Reviewer #1

(Remarks to the Author)

The authors were responsive to my critiques by acknowledging the study's limitations and appropriately tempering the tone of their conclusions. While no additional conclusive data were provided to fully address my concerns, the revisions have satisfactorily improved the manuscript. At this stage, I have no further concerns.

(Remarks on code availability)

First, we would like to thank the editorial board and the reviewers for the in-depth review of our manuscript and the many constructive comments.

In response to these comments, we have made extensive revisions that we believe substantially strengthen the manuscript. To summarize, key additions include:

1. Extensive immunofluorescence stainings validating proximal tubule (PT) and thick ascending limb (TAL) injury clusters in both mouse and human samples.
2. Immunofluorescence stainings confirming the spatial proximity of injured PT cells to macrophages, fibroblasts and T cells in mouse and human samples.
3. Results from unbiased spatial domain identification.
4. Single-cell deconvolution analysis of more than 3,800 kidney transplant biopsies (including 298 TCMR biopsies) with microarray data, demonstrating enrichment and specificity of injury-associated cell states in TCMR.
5. A systematic comparison of gene expression changes in TCMR between indication and protocol biopsies.
6. Fully revised main figures and many new supplemental figures supporting and investigating the correspondence between:
 - a. Xenium and single-cell transcriptomic data and (sub) cell type labels.
 - b. mouse and human TCMR data.

All these points resulted in substantial revisions of the originally provided figures and supplement, the addition of 2 new main figures and 17 supplemental figures. We believe that these results further substantiate our original findings. All content-relevant additions to the manuscript are highlighted in red font.

Please note that we needed to substantially shorten and fully re-write the abstract (previously 242 words, initially accepted) to comply with the journal's formatting requirement of 150 words or less. The content remained unchanged, except for the inclusion of additional experiments and analyses conducted during the revision.

Please find our point-by-point responses to the specific comments below:

Reviewer #1 (Remarks to the Author):

The manuscript focusses on the evaluation of TCMR argument that the molecular pathways triggered by TCMR remain poorly understood. The authors state that although when TCMR is treatable, it associates with reduced allograft survival, suggesting current therapies are not adequately addressing all the aspects of TCMR-induced injury. This statement represents the

scientific premise of the study. They evaluate using single cell approaches (snRNAseq and spatial transcriptomics) kidney tissues from two mouse models of TCMR (including allo- and iso-genic kidney transplant mouse models). Although the authors claim that the study aim is to delineate the molecular changes associated with TCMR and evaluate their impact on allograft outcomes, they mainly focused on the analyses of the epithelium response to TCMR-associated injury. Nevertheless, this is the more novel aspect of the study. They compared the findings from mouse data to molecular data from human biopsies. Finally, they use bulk transcriptomics from a large cohort of patients to assess potential clinical relevance for elucidating potential therapeutic targets. The study includes allo- and iso-genic kidney transplant mouse models. Overall, the manuscript is well presented, organized, and test an important hypothesis. The use of single cell approaches, interspecies comparison, and validation using bulk transcriptomics is a strength of the study. Focus on the response of the epithelium to the injury is also relevant. The study effectively links mouse and human datasets, demonstrating conserved injury states in PT and TAL cells.

Reply: We thank the reviewer for this favorable assessment of our manuscript.

However, some points that deserve further evaluation were identified including:

1- Most of the single cell transcriptomics evaluations seems superficially evaluated. Although cell identity was established for PT subclusters (PT injurym1-4 and PT proliferating), the results about specifics of these clusters are limited. There are no specific results defining these cells subclusters and/or analyses about characterization of cell type-specific pathways. The description of the PT injury m4 cell cluster (identified as most injured cluster by number of injury-associated marker genes), is limited to “The marker genes for PT Injury m4 include several associated with maladaptive repair cells in AKI, such as Vcam1, Cd44 and Vim22,24”. Overall, most the results rely on markers previously described in the literature for native kidneys with AKI. Likewise, the same limitations affect the evaluation of TAL cell subclusters.

Reply: To address this important comment, we now include and discuss relevant marker genes of these clusters in the respective section (section “Mouse TCMR elicits spatially diverse injured cell states in PT and TAL”). We also adapted the presentation of marker genes, which now convincingly shows the overlap with the human marker genes and the Xenium data (Fig. 3C, H (mouse) versus Fig. 6C, F (human) versus Suppl. Figs. S9B and 11B (Xenium)).

2- Although the main interest seems to focus on PT and TAL cell clusters, immune cells and epithelium-immune interactions are critical as part of resolution of injury or progression to impaired repair during TCMR. However, the cell type-specific characterization of immune cells is minimal. Specifically, what type of macrophages were involved in injured PT-immune cell interactions? Fig 4 A shows a subcluster analyses of immune cells with 4 macrophages

populations. However, no characterization of the macrophage subpopulations is provided nor their associated profiles. Same point is valid for all the immune cells.

Reply: We thank the reviewer for this important comment. We characterized and re-annotated the identified immune cell subclusters in the mouse and the human data using cell type-specific markers from the literature¹⁻⁴ (see Fig. 4 and new Suppl. Fig. 20). Specifically, we were able to distinguish between CD4+ T helper cells, CD8+ cytotoxic T cells, regulatory T cells, B cells and NK cells within the lymphoid cells in both the mouse and the human data as well as a plasmacytoid dendritic cell population in the human data. Within the myeloid cells, we could differentiate between dendritic cells and monocytes as well as resident, infiltrating and activated/ lipid-associated macrophages, although some marker genes were expressed in more than one macrophage population indicating some transitional macrophages within the identified clusters. Resident macrophages were for example marked by expression of *Mrc1* and lipid associated macrophages by expression of *Trem2*, *Spp1* as well as genes encoding proteins involved in lipid handling such as *Fabp4* and *Fabp5*. The “Activated Macrophage” population showed expression of profibrotic genes such as *Col1a1* and other collagen encoding genes, such as *Sparc* and *Cxcl12*. The cluster termed “Infiltrating Macrophage” retained slight expression of some monocyte-characteristic genes, but predominantly showed expression of macrophage markers without a specific transcriptomic profile, possibly also indicating a transitional state.

Following a comment from Reviewer 3, we re-analyzed spatial proximity of injured PT and TAL clusters using a published tool. This analysis confirmed the previously observed depletion of all broad leukocyte cell types in the vicinity of TAL Injury m3 and an increased presence around PT Injury m4 (Fig. 4E). We also performed this analysis for the leukocyte subtypes shown in new Suppl. Fig. S16. Especially activated macrophages are highly overrepresented in the vicinity of PT Injury m4. This finding is now also included in the main body (section “TCMR-induced injured epithelial cell states show heterogeneous proximity to immune cells and surrounding cell type environments”).

3- In the cross-species evaluation, the study effectively bridges mouse and human datasets, demonstrating conserved injury states in PT and TAL cells. However, although injured PT and TAL cells were identified, their relative abundance was not significantly higher in TCMR samples compared to stable allografts. How alike were the PT injured specific profiles between conditions? This point deserves further evaluation/discussion.

Reply: We thank the reviewer for raising this point. We acknowledge that the abundances of injured epithelia were not significantly increased in TCMR versus no TCMR. This can result from several expected factors, but primarily since the abundances of these populations are not constant over time in all patients. As we do not exactly know (in contrast to the mouse model)

at which time point after TCMR onset the biopsies were taken, this might explain some variability among patients (see also Fig. 8B).

To address this important reviewer comment, we have therefore performed and now include the following analyses:

1. New single-cell deconvolution analysis of microarray data from 3858 biopsies obtained from 3210 kidney transplant recipients including 298 TCMR biopsies, enabling us to meaningfully compare abundances between conditions, which shows:
 - a. Significantly increased abundances of the most severely injured cell states in PT and TAL (PT Injury h2 and TAL Injury h2) (see Fig. 6I) and
 - b. reduced abundances of PT Injury h1 and TAL Injury h1 indicating that these injury states might originate from other sources of kidney transplant injury (Fig. 6I).
 - c. A systematic comparison of gene expression changes showing - strong and significant correlations – between mouse and human snRNA-seq TCMR data as well as human TCMR microarray data (Suppl. Fig. S21).
2. Fully updated marker gene heatmaps better highlighting the correspondence of injury clusters between mouse and human (compare heatmaps in Figs. 3 and 6).
3. Immunofluorescence validation of critical markers for injured PT and TAL clusters in mouse and human (Figs. 3 E, J and 7, Suppl. Figs. 9, 10 and 12).

We hope these new analyses more convincingly demonstrate the correspondence between mouse and human TCMR data, as well as the induction of severely injured PT and TAL states by TCMR in human.

4- The human biopsies were obtained from protocol biopsy program at Hannover Medical School, meaning they were not necessarily taken at the time of acute rejection symptoms but rather as part of scheduled monitoring of kidney transplant patients. This contrasts with the controlled timing in mouse models, reducing direct comparability. This is likely reflected by the absence of the cluster of proliferating cells in the human samples. Further discussion about how this issue may affect other cell clusters and moreover, their specific transcriptome must be discussed. Also, this reviewer was not able to identify the Banff scoring for human samples with TCMR used in the snRNAseq study or the specific time of biopsy collection.

Reply: Thank you for this comment. To address it, we systematically compared TCMR protocol and indication biopsies in our microarray dataset and observed extremely high correlations in gene expression. These results are now included in Suppl. Fig. S21A and are mentioned in the results section (section “Human TCMR kidney allografts exhibit injured cell states resembling those in mice”). In summary, our analyses do not indicate major differences between protocol and indication biopsies in TCMR.

We agree with the reviewer that the absence of the proliferating cluster is most likely attributable to the timing of the human biopsies, a phenomenon also observed in AKI and CKD samples. We do not consider this to affect the interpretation of the other cell types of interest in this manuscript as supported by the parallel analyses in mouse and human datasets.

5- The study focused primarily on PT and TAL injury states, but other cell types (e.g., immune cells, endothelial cells, fibroblasts) may contribute significantly to TCMR pathogenesis. Additional cell type-specific analyses could enhance the understanding of injury mechanisms in response to TCMR.

Reply: Thank you for this important remark. Indeed, we focused on PT and TAL cells, as these cell types exhibit the most pronounced response to TCMR—both in terms of differential gene expression and the abundance of TCMR-induced injury clusters.

Wherever applicable, we applied the same analytical approaches across all cell types, including differential gene expression, pathway enrichment and subclustering analyses.

In the revised manuscript, we have now included unbiased domain analyses (new Fig. 5) that encompasses all cell types. This analysis revealed, for example, that the severely injured PT Injury m4 cells predominantly reside in inflammatory microenvironments surrounded by immune cells and fibroblasts. We validated the presence of these cell groups using our Xenium data as well as new immunofluorescence stainings in mouse and human (see Figs. 5 and 7 as well as Suppl. Fig. S18).

We hope that these additional data satisfactorily address the reviewer's comment. While we acknowledge that many more analyses - particularly involving other cell types—could be performed, we have intentionally kept the focus of the manuscript on epithelial injury impact and patterns. We hope the reviewer agrees with this approach.

6- The longitudinal analysis included only 12 patients, which is a small sample size to draw definitive conclusions about persistence of injury states post-TCMR.

Reply: We are aware of this limitation and had highlighted this in the discussion section. We agree that the sample size is small, but it interestingly still displays that some patients keep signatures of injured epithelia. Certainly, larger cohorts would enable to identify the amount and conditions under which patients would display such a phenotype. This, of course, cannot be answered by our small cohort. As we find the plots in Fig. 8B (previously Fig. 6B) still to be intriguing, we kept them in the figure and hope that this is acceptable for the reviewer.

7- The significant variability in injury-related gene set scores among TCMR patients indicates heterogeneous injury responses, making it challenging to define a universal injury signature. This suggests that additional patient-specific factors (e.g., immune responses, comorbidities,

treatment regimens) may influence epithelial injury progression and should be further investigated.

Reply: Thank you for this important remark. We included these limitations in the discussion.

8- Although TAL injury signatures were more pronounced in TCMR, they were not exclusive to it, as they also correlated with injury in AKI. This suggests that while TAL injury is relevant for TCMR outcomes, other forms of allograft stress (e.g., ischemia) may also contribute to its expression.

Reply: Thank you for raising this important point. In fact, it is unknown but not unlikely that other conditions, which trigger epithelial injury, might also entail this kind of TAL injury. In fact, together with our new data from deconvolutional analysis, we can now state that:

1. The mere presence of the discovered injury cell states affects different conditions after kidney transplantation alike, e.g. ABMR, TCMR (Fig. 8, Suppl. Fig. S26).
2. The abundance of certain injury cell states, namely PT Injury h2 and TAL Injury h2, is significantly increased during TCMR (when compared to no rejection or ABMR, Fig. 6I).

We thank the reviewer for prompting us to clarify this important point.

Reviewer #2 (Remarks to the Author):

T-cell-mediated rejection (TCMR) remains a significant hinderance to long-term kidney transplant survival, as it contributes to graft injury and eventual failure despite immunosuppression strategies. To better understand molecular mechanisms associated with TCMR the authors induced acute TCMR in mouse kidney transplant models and analyzed molecular changes using snRNA sequencing and spatial transcriptomics. These findings were compared with human biopsy data from rejected and stable allografts. The study revealed that TCMR causes significant gene expression changes and epithelial injury, particularly in proximal tubules (PT) and thick ascending limbs (TAL), with spatial heterogeneity and proximity to immune cells. Cross-species analysis confirmed similar injury patterns in human TCMR cases. Furthermore, kidney transplant outcomes were linked to the persistence of injured epithelial states, even after rejection resolution. The manuscript reports interesting data with potential value in expanding current understanding about TCMR and underlying molecular landscapes. The results are significant despite the small sample size in the validation cohort.

Reply: We thank the reviewer for this positive assessment of our manuscript.

There are a few aspects the authors could do to improve the quality of the manuscript's claims and conclusions and make it more useful to its readers. The primary concern about the

manuscript is that the conclusion of this study is inconclusive in terms of the utility of such gene set scores in enabling clinicians to triage transplant recipients into different bins of TCMR.

Reply: We thank the reviewer for this important comment. Based on our data, we propose that the epithelial injury signatures presented here should be monitored and evaluated in TCMR patients during follow-up visits. We have previously demonstrated the general feasibility of quantifying epithelial cell states in urine in the context of AKI⁵. Applying similar approaches in the context of TCMR may enable prospective correlation of these epithelial states with clinical outcomes.

Currently, no therapies exist to specifically target injured PT and TAL states. Thus, further research is needed to develop interventions aimed at these cell states or the mechanisms that sustain them, and to assess their impact on graft outcomes.

In summary, if these epithelial cell states can be validated as predictive markers of allograft prognosis, and if targeted therapies become available, patients experiencing TCMR could potentially be stratified according to their risk profile and treated in a personalized manner. As these considerations pertain primarily to future directions, they were mentioned in the discussion section.

The sample size is small, and despite the molecular score based on the snRNA seq data from mice and humans, other injury types in transplant have not been discussed at all. The authors have brought AKI and CKD studies, but have failed to discuss how to interpret the data in the context of ABMR and other transplant injuries such as mixed rejection.

Reply: Thank you for this comment. We have updated the manuscript accordingly and now include:

1. Kaplan Meier curves for ABMR and mixed rejection (see Fig. 8A and Suppl. Fig. S26) and
2. single-cell deconvolution analyses from microarray data on 3858 kidney biopsies investigating abundances of injured cell states in these data (Fig. 6I).

In summary, we can now show that:

1. The mere presence of the discovered injury cell states affects different conditions after kidney transplantation alike, e.g. ABMR, TCMR or mixed (Fig. 8A, Suppl. Fig. S26).
2. The abundance of certain injury cell states, namely PT Injury h2 and TAL Injury h2, is significantly increased during TCMR when compared to e.g. ABMR or no rejection (Fig. 6I).

We also included these new results in the revised discussion.

Minor:

For Fig 1D, top panel, please provide color codes for major cell types.

Reply: We apologize, the color code is identical to the color code of the UMAP. The figure legend was modified accordingly to better emphasize this.

Fig 6A-- How would other phenotypes behave other than TCMR and nonTCMR in Kaplan Meier curves? Did nonTCMR cohort include ABMR? How did the gene set score perform for mixed rejection cases?

Reply: As stated above, we have now included analogous curves for the other subgroups in the paper (see. Suppl. Fig. S26). The initial nonTCMR cohort included patients with ABMR. These cohorts were now separated for more clarity (compare Fig. 8A, Suppl. Fig. S26).

Fig 6B-- There is no mentioning of the significance of orange vs grey circles in the figure legend or the result section where the data is presented. Please provide the information both in the legend as well as the result section.

Reply: We apologize, this is now included.

Reviewer #3 (Remarks to the Author):

In this manuscript, the authors employ a murine kidney transplantation model to investigate epithelial cell injury states during acute T cell-mediated rejection (TCMR). They performed snRNA-seq on kidneys from both syngeneic and allogeneic transplant recipients (BALB/c and C57BL/6), identifying distinct injury-associated subpopulations within proximal tubule (PT) and thick ascending limb (TAL) cells. Spatial transcriptomics (Xenium) was used to characterize immune cell subtypes within the rejecting allografts. In addition, the authors derived injury gene expression signatures from snRNA-seq analysis of human kidney transplant biopsies and correlated these injury-associated signatures with reduced graft survival in an independent bulk transcriptomic dataset.

The core hypothesis—that injured epithelial cell states contribute to allograft dysfunction—is conceptually important, and the integration of scRNA-seq with spatial technologies is a valuable strategy. However, this manuscript fails to meet the analytical and experimental standards necessary to support such claims. The analysis lacks rigor, fails to provide cell type-specific insights, and omits essential validation steps. Several figures are descriptive but not interpretable in their current form. I do not believe the manuscript is suitable for publication. Substantial re-analysis, validation, and clarification are needed to bring the study to the level expected by the field. Below are my specific comments.

Reply: We thank the reviewer for the thorough and detailed review of our manuscript. We have carefully considered all comments and hope we have addressed each of the points raised, as outlined below.

General remark from the authors:

We thank the reviewer for highlighting the importance of reliable cell type annotation in spatial datasets, a point we fully agree with and consider essential. As this concern was raised multiple times across the reviewer's comments, we would like to present our general perspective here to avoid redundancy in the point-by-point responses that follow.

We agree that confirming the presence of key marker genes, providing diagnostic evidence such as expected anatomical localization and validating selected cell populations are crucial for spatial transcriptomics datasets. In response, we have further strengthened these aspects throughout the revised manuscript (please refer to the specific responses below for details).

The reviewer requested to validate clustering results from Xenium data by directly comparing them to snRNA-seq-derived clusters. While some sub cell types, such as within PT and TAL, can be recapitulated from unsupervised clustering in both data modalities, other compartments (notably leukocytes) do not resolve into matching subclusters in the Xenium data.

It is expected that clustering of spatial transcriptomic data does not necessarily reproduce all subclusters obtained from single-cell or single-nucleus RNA sequencing, since these complementary approaches are based on different techniques (imaging versus sequencing-based), target a different number of genes (479 genes in the Xenium panel in this study versus whole transcriptome) and result a different gene coverage (hundreds compared to thousands of transcripts per cell)⁶⁻⁸. Reference-based annotation approaches, such as RCTD or SingleR, have therefore become widely used in spatial transcriptomics, precisely because they leverage the transcriptome-wide coverage and sequencing depth of scRNA-seq to annotate spatial datasets more reliably.

To further illustrate this point, we have performed several additional analyses for the reviewer validating our label transfer approach. Specifically, for leukocytes, we downsampled our high-resolution snRNA-seq data to match both the gene panel and UMI distributions of the Xenium dataset (see RevFig. 1). When subjected to unsupervised clustering, both, the downsampled snRNA-seq and the original Xenium leukocyte data separated into similar clusters corresponding to broad leukocyte cell types (e.g. T cells, myeloid cells), while finer subdivisions were not clearly recovered (see RevFig. 1A). Applying label transfer to the downsampled snRNA-seq data produced however meaningful sub cell type annotations (RevFig. 1B). This supports the use of reference-based approaches, even when clustering resolution is limited and enabled sub annotation of leukocytes, a particularly challenging population to annotate due to its location between multiple other cell types in the tight interstitial space.

In this way, whole transcriptome-based sequencing approaches and high-resolution imaging-based spatial transcriptomic approaches can complement each other, but we, of course, agree that additional secondary validation using diagnostic plots and protein-based techniques such as immunofluorescence validation can further support conclusions from the transcriptomic-based approaches.

We took the reviewers concerns very seriously and included additional marker gene expression analyses, diagnostic spatial plots and validation experiments to support our annotations, all of which are detailed in the point-by-point responses and the revised manuscript. In fact, we devoted a large portion of the new supplementary material to these concerns (Suppl. Figs. S5, S8-12, S18, S22). We re-annotated all Xenium data by snRNA-seq label transfer using the SingleR package, as it is simpler to implement and interpret and has recently been shown to achieve superior performance for Xenium cell type annotation. In addition, during revision, we noticed that technical support for RCTD is very limited. As depicted in the figures, cell type annotation did not differ by much between the tools.

Reviewer Figure 1: Comparison of unbiased clustering results in leukocytes from downsampled snRNA-seq and Xenium data. A. Gene expression from snRNA-seq leukocytes was downsampled to the genes included in the Xenium panel and to comparable sequencing depth (see histograms). Unbiased clustering identified major leukocyte cell types but did not resolve all subtypes. **B.** SingleR label transfer applied to downsampled snRNA-seq data (columns, known cell type) enabled refined annotation of leukocyte (sub) cell types.

1. In figure 2A, if the BALB/c to C57BL/6 kidneys has milder rejection, why there are more DEGs in PC, EC, intC, PEC and IC-A when comparing to the C57BL/6 to BALB/c kidneys?

Reply: Thank you for this remark. Indeed, rejection and injury in the C57BL/6 to BALB/c model are more severe, as reflected by higher expression of injury markers (snRNA-seq, ST and immunofluorescence), increased mortality and Banff scoring. We therefore believe that referring to C57BL/6 to BALB/c as the harsher rejection model is justified. Additionally, differentially upregulated genes, for example in the CD-PC, were enriched for metabolic pathways and may indicate strain-specific differences in TCMR. All DE genes are provided in the supplemental tables. We acknowledge these findings in the main text but do not elaborate further, as they are beyond the scope of this study.

2. The heatmap shown in Figure 2B provides minimal information beyond the number of DEGs in each cell type. The authors should highlight key cell type-specific genes, particularly in PT and TAL cells, to improve interpretability. In addition, experimental validation of select disease-associated genes in these cell types would considerably strengthen the study's conclusions.

Reply: Thank you for this remark. Indeed, much of the gene expression changes are, as expected, observable across cell types which is likely induced by the cytokine environment and general (epithelial) responses to injury.

Following the reviewer's suggestion, we moved the previous Fig. 2B to the supplement (Suppl. Fig. S6B) and now highlight key examples of deregulated genes between rejection and control in the main figure and in the supplement (Fig. 2B-E, Suppl. Fig. Suppl. Fig. 7D). Most of these genes could be validated using our Xenium data. We analyzed all DEG modules identified by unbiased clustering using Weighted Gene Correlation Network Analysis (WGCNA) and provide the top deregulated genes specific to PT and TAL (Suppl. Fig. S7). As stated in the main text (section "Mouse TCMR induces pronounced gene expression responses in kidney epithelial cells"), the overall pattern of enriched pathways across cell types is reflected in the top DEGs of individual cell types, although some genes, including injury markers and cytokines, show clear cell type-specific expression.

3. Figure 2C basically shows that all these cell types share most of the disease pathways. Therefore, the authors should conclude that few cell type-specific pathways were identified, which makes the overall conclusion less compelling. The authors should focus more on highlighting any truly cell type-specific disease pathways to strengthen the impact of their findings.

Reply: Thank you again for these remarks to Figure 2. We tried to address this in the revised version of the manuscript as stated in reviewer comment #2.

4. In Figure 3A, PT injury m2–3 cannot be classified as distinct injured states, as they share all marker gene expression and lack unique disease genes. If the authors want to define these as transitional injury states, pseudotime trajectory analysis would be more appropriate than clustering analysis.

Reply: Thank you for raising this point. In the revised figure we included additional marker genes to demonstrate that PT Injury m2 and m3 are indeed distinct clusters (Fig. 3C), analogous for TAL (Fig. 3H). In addition, pseudotime analyses confirmed that they progress towards the most severely injured states, PT Injury m4 for PT and TAL Injury m3 for TAL (Fig. 3A, F). Please note that we now also include overlapping marker gene expression in the Xenium data to support validity of the label transfer for PT and TAL (Suppl. Figs. S8B and S11B).

5. The accuracy of mapping scRNA-seq to Xenium data depends heavily on the gene panel used in Xenium—for example, whether markers for PT subtypes are included. As a result, it is difficult to assess the accuracy of the spatial cell mapping shown in Figure 3A. The authors should apply the same mapping algorithm to PT S1–S3 subtypes to determine whether PT S3 localizes to the corticomedullary junction and PT S1/2 to the outer cortex, as expected. This can serve as internal validation of the approach.

Reply: This is indeed a critical point. As suggested by the reviewer, we performed the same mapping for the healthy PT and TAL segments. This resulted in the expected localization and is now included in Suppl. Figs. S8 (for the PT) and S11 (for the TAL).

6. In Figure 3B, if PT injury m1 represents an injured cell type, it is unclear how to explain the lack of change in its proportion between allogeneic and syngeneic kidneys. The only marker used to define PT injury m1 is *Nqo1*. Since this is a novel marker in this context, the authors should provide additional information about the gene, including its primary function and potential role in tubular injury. Furthermore, immunofluorescent staining is needed to confirm the presence of this injury state in the allogeneic kidney.

Reply: Thank you for this important remark. Upon reinvestigation of the clustering, we found that the previous PT Injury m1 cluster comprised two biologically distinct subclusters, one corresponding to healthy PT S3 medullary cells and the other to injured PT cells. Following the reviewer's comment, we refined the clustering of PT Injury m1 (Fig. 3A-D). We also provide validation for the existence of this cluster through additional immunofluorescence stainings and Xenium data (Suppl. Figs. S9 and S10). Please note that *Nqo1* is no longer a marker of the injured fraction of the revised PT Injury m1 cluster and is now correctly assigned as a marker of PT S3 medullary cells (Fig. 3C).

Please do also note that this new clustering slightly alters downstream analyses including correspondence of injury clusters between mouse and human (Fig. 6).

7. All of the concerns raised above regarding the PT subtypes (points #4–#6) also apply to the TAL subtypes shown in Figure 3C and 3D.

Reply: We tried to address all points for PT and TAL in points #4–#6, including pseudotime analysis, expression of marker genes and spatial localization of clusters.

We noticed that some clusters previously annotated as cTAL, specifically cTAL2 and cTAL3, also extend to some degree into medullary regions. We therefore renamed them as cmTAL1 and cmTAL2 (cm = corticomedullary). We hope the reviewer agrees that this revised annotation more accurately reflects their localization.

We also noted a plotting error in the previous Fig. 3C, where TAL Injury m1 was not sufficiently displayed due to overplotting. We apologize for this.

8. Since the authors have generated Xenium data, they should perform subclustering analysis on the PT and TAL populations within the Xenium dataset to validate the injured cell states and assess their spatial distribution in allogeneic versus syngeneic kidneys.

Reply: We thank the reviewer for this valuable suggestion. In response, we have performed subclustering analyses of the PT and TAL populations within the Xenium dataset. This analysis revealed comparable subclusters as discovered in the snRNA-seq data (RevFigs. 2 and 3). Small differences were noted in cell abundances, for example clustering overestimated TAL Injury m3 abundance compared to immunofluorescence stainings which aligned more closely with annotations generated from label transfer. In contrast, spatial distribution was highly comparable. Moreover, critical findings such as spatial localization of PT Injury m1, PT Injury m4 and TAL Injury m3 were validated using our Xenium data and/or immunofluorescence stainings which confirmed our assignment of PT and TAL subclusters in the Xenium data from label transfer (Fig. 3E, J, Suppl. Figs. 9, 10 and 12).

The clustering approach of ST data was not able to differentiate between the different healthy TAL clusters, e.g. cTAL1-2, cmTAL1-2 and mTAL1-2. However, as shown by marker gene expression and spatial expression domains, these clusters seem to exist (see Fig. 3 and Suppl. Fig. S11, presence of e.g. *Cldn16* in cTAL1 and cmTAL1 while absent or low in cTAL2 and cmTAL2, vice versa for *Cldn10*).

Reviewer Figure 2: Comparison of clustering and label transfer for Xenium PT cells. A. Left plot: UMAP from unbiased clustering of Xenium PT cells using highly variable genes. Right plot: Confusion matrix showing overlap between cell type assignment from clustering and from label transfer. **B.** Spatial distribution of healthy PT clusters S1/2, S3 and S3 medullary based

on unbiased clustering (left) or from snRNA-seq label transfer (right). **C.** Spatial distribution of injured PT clusters based on unbiased clustering (left) or from snRNA-seq label transfer (right).

Reviewer Figure 3: Comparison of clustering and label transfer for Xenium TAL cells. A.

Left plot: UMAP from unbiased clustering of Xenium TAL cells using highly variable genes. Right plot: Confusion matrix showing overlap between cell type assignment from clustering and from label transfer. Note that cTAL1/2 and cmTAL1/2 were merged into a single cTAL group to match the clustering resolution. Similarly, mTAL1/2 were combined into mTAL. B. Spatial distribution of healthy TAL clusters cTAL – cortical TAL and mTAL – medullary TAL based on unbiased clustering (left) or from snRNA-seq label transfer (right). C. Spatial distribution of injured TAL clusters based on unbiased clustering (left) or from snRNA-seq label transfer (right).

9. *It is unclear why the authors focus exclusively on immune cells in Figure 4 for the Xenium data analysis. A more comprehensive analysis should include all cell types to determine whether the same populations identified by scRNA-seq can also be resolved using Xenium data alone. In Supplementary Figure S4A, the authors perform clustering on the Xenium dataset, but substantially fewer cell types are shown, and no subclustering analysis was conducted for PT and TAL.*

Reply: In Figure 4, we focused on immune cells as they represent a pivotal cell population during rejection. They do not only represent a histological hallmark (interstitial nephritis and tubulitis), but their impact is also intensively discussed in the literature. For us, it was hence an intuitive question to ask which spatial proximity exists between different immune cell populations and injured epithelial cell states. We agree that a more comprehensive spatial proximity analysis can be informative which we tried to address with an unbiased spatial domain clustering approach now included in the revised manuscript (new Fig. 5).

Regarding the discovery of cell types observed in the single cell data when compared to the Xenium data, we kindly refer to our general remark at the beginning of this reply to the reviewer, including RevFig. 1 and the response to comment #8.

10. *The authors need to provide details on how the 100 custom genes were selected for their Xenium gene panel design. For instance, if the panel lacks sufficient markers for injured PT or TAL subtypes, the entire study would be compromised. In such a case, the key findings from the scRNA-seq data regarding injured PT and TAL cells would be difficult to validate or reproduce using the Xenium platform, making integration between the two datasets unreliable. Furthermore, the absence of appropriate markers would preclude meaningful downstream analyses, such as determining the spatial roles of injured PT and TAL subpopulations. For example, niche analysis—which could reveal which injured states preferentially recruit immune cells—would not be possible, thereby limiting mechanistic insights into tubular injury during TCMR.*

Reply: Thank you for pointing this out. The Xenium panel consisted, as pointed out in the methods section, of 479 genes of which 379 originated from a pre-designed mouse multi tissue panel (Mouse Tissue Atlas v1) and 100 custom genes. The 100 custom genes were designed based on previous single cell experiments on mouse kidneys in the context of acute kidney injury and rejection. Therefore, these genes included canonical marker genes for all major kidney cell types and expected injury-induced cell populations. We furthermore added a sentence to the methods section commenting on the origin and selection of the 100 additional genes.

11. It is unclear how the authors generated Figure 4D, as they did not perform subclustering of PT and TAL cells. It appears that computational methods were used to map snRNA-defined cell types onto the spatial data. however, this mapping relies on the set of genes shared between the snRNA-seq and Xenium platforms. If the gene panel design lacks sufficient markers for injured states, particularly for PT and TAL, the resulting cell mapping may be highly inaccurate.

Reply: Thank you for this comment. Indeed, Figure 4D (now Fig. 4E) was created by annotating spatial transcriptomics data with clustering results and labels from snRNA-seq data. We fully agree with the reviewer that overlapping marker gene presence in the Xenium panel is required for the correct annotation of the spatial data. As pointed out above, we highlighted and presented marker genes present in both modalities (snRNA-seq and Xenium) (Fig. 3, Suppl. Figs. S8 and S11).

12. In addition, for the neighborhood analysis in Figure 4D, the authors simply report the percentage of neighboring PT and TAL cells, which is a descriptive rather than statistical approach. The authors should utilize publicly available tools designed for spatial neighborhood analysis to reanalyze the data more rigorously. To support their findings, they should also provide immunostaining data to confirm that the identified cell types are indeed in close spatial proximity.

Reply: Thank you for this comment. We completely revised our spatial proximity approach and used, as suggested by the reviewer, publicly available tools (co-occurrence probabilities as implemented by squidpy), which confirmed our previous results of co-localization of specific injury cell types with immune cells (see revised Fig. 4E, Suppl. Figs. S15-17). This analysis confirmed the enrichment of T cells and macrophages in proximity to PT Injury m4 cells, while they were depleted in proximity of TAL Injury m3 cells.

Additional to the direct neighboring cell analysis in Fig. 4E (previously 4D), we now performed an unbiased spatial neighborhood analysis and immunostaining validation. We devoted an entirely new Figure to these analyses (new Fig. 5). Please note that the identification of spatial

domains did not rely on any cell type labeling from our side but was merely based on Xenium gene expression. Apart from the fact that this gene expression-driven analysis identified the expected compartments (renal medulla, glomeruli etc.) with the expected cell types (based on label transfer), we were able to identify microenvironments comprised of heavily injured PT cells (mainly PT Injury m4) in proximity to immune cells (macrophages and T cells) as well as activated fibroblasts. We validated this in mice and human by immunostaining (Figs. 5 and 7, Suppl. Figs. 18 and 22).

13. In Figure 5, the authors focus solely on demonstrating the similarity of cell types between human and mouse in TCMR. However, they fail to present more critical data—specifically, whether the gene expression changes observed between stable allograft and TCMR biopsies in humans are similar to, or distinct from, those observed between allogeneic and syngeneic kidneys in the mouse model. This comparison is essential to establish the translational relevance of the mouse findings to human disease.

Reply: We thank the reviewer for suggesting this important validation. In response, we now provide several new results further strengthening the reliability of the translational interpretation:

1. Fully updated marker gene heatmaps, indicating overlap between mouse and human snRNA-seq (Figs. 3 and 6).
2. Immunofluorescence validation of key PT and TAL injury marker genes with immunofluorescence in, both, mouse and human samples (Fig. 3, 7 and Suppl. Figs. 9, 10, 12).
3. Systematic comparison of mouse and human snRNA-seq data with a large cohort of human TCMR microarray samples supporting strong correlation between mouse and human but also with a much larger gene expression cohort (298 TCMR biopsies) (Suppl. Fig. S21).

All three analyses now confirm strong similarities between mouse and human snRNA-seq data and are in line with results from an additional, much larger TCMR cohort.

14. Since the authors selected different sets of genes in Figure 5C and 5F compared to those used in Figure 3A and 3C, it is unclear whether the human injured cell states truly correspond to the mouse injured cell states. The simple Pearson correlation analysis presented in Figure 5F and 5H is insufficient to support the conclusion that these are the same cell types. Additional analyses are needed to strengthen this claim. For example, the authors should identify injury markers shared between human and mouse tubular subtypes and perform immunostaining to validate the identity of these matched cell states.

Reply: Thank you for this comment. As stated above (e.g. reply to reviewer comment #13), we updated the heatmaps to reflect shared marker genes across species (Fig. 3C, H (mouse), Fig. 6C, F (human)) and validated PT and TAL injury population markers by immunofluorescence staining (Figs. 3 and 7, Suppl. Figs. S9, S10 and S12). We also replaced the initial Spearman correlation analysis with a published label transfer approach using SingleR (Fig. 6E, H).

15. The authors should use the gene signatures derived from the human biopsy scRNA-seq data to perform cell type deconvolution on the bulk transcriptomic dataset presented in Figure 6. This analysis would allow them to determine whether the proportions of injured tubular subtypes are specifically increased in TCMR or mixed rejection biopsies, but not in ABMR biopsies. Such evidence would provide stronger support for the relevance of the identified injury signatures in distinguishing rejection types.

Reply: Thank you for this important comment. As our human snRNA-seq dataset is limited in size, we performed single-cell deconvolution on a large kidney transplant cohort comprising 3858 biopsies from 3210 patients (2155 without rejection, 1107 with ABMR, 298 with TCMR, and 298 with mixed rejection) (Fig. 6I).

Briefly, we observed the expected depletion of healthy epithelia and a significant overrepresentation of PT Injury h2 and TAL Injury h2 in TCMR and mixed (TCMR + ABMR) samples. Both clusters were also significantly more abundant in TCMR than in ABMR. Interestingly, PT Injury h1 and TAL Injury h1 were depleted across most rejection groups, suggesting a potentially different origin of these injured states.

We would like to emphasize that, although PT Injury h2 and TAL Injury h2 are enriched in TCMR, we never claimed that they ultimately can distinguish between all rejection types. Rather, we stated that *i)* TCMR induces these injured cell states, *ii)* they are clinically relevant but currently not therapeutically targeted and *iii)* they persist in some patients despite successful rejection treatment.

References:

- 1 Conway, B. R. *et al.* Kidney Single-Cell Atlas Reveals Myeloid Heterogeneity in Progression and Regression of Kidney Disease. *J Am Soc Nephrol* **31**, 2833-2854, doi:10.1681/ASN.2020060806 (2020).
- 2 Jaitin, D. A. *et al.* Lipid-Associated Macrophages Control Metabolic Homeostasis in a Trem2-Dependent Manner. *Cell* **178**, 686-698 e614, doi:10.1016/j.cell.2019.05.054 (2019).
- 3 Reck, M. *et al.* Multiomic analysis of human kidney disease identifies a tractable inflammatory and pro-fibrotic tubular cell phenotype. *Nat Commun* **16**, 4745, doi:10.1038/s41467-025-59997-4 (2025).
- 4 Vegting, Y. *et al.* Infiltrative classical monocyte-derived and SPP1 lipid-associated macrophages mediate inflammation and fibrosis in ANCA-associated glomerulonephritis. *Nephrol Dial Transplant* **40**, 1416-1427, doi:10.1093/ndt/gfae292 (2025).

- 5 Klocke, J. *et al.* Urinary single-cell sequencing captures kidney injury and repair processes in human acute kidney injury. *Kidney Int* **102**, 1359-1370, doi:10.1016/j.kint.2022.07.032 (2022).
- 6 Cheng, J., Jin, X., Smyth, G. K. & Chen, Y. Benchmarking cell type annotation methods for 10x Xenium spatial transcriptomics data. *BMC Bioinformatics* **26**, 22, doi:10.1186/s12859-025-06044-0 (2025).
- 7 Janesick, A. *et al.* High resolution mapping of the tumor microenvironment using integrated single-cell, spatial and in situ analysis. *Nat Commun* **14**, 8353, doi:10.1038/s41467-023-43458-x (2023).
- 8 Cable, D. M. *et al.* Robust decomposition of cell type mixtures in spatial transcriptomics. *Nat Biotechnol* **40**, 517-526, doi:10.1038/s41587-021-00830-w (2022).

We thank the editor and the reviewers for their very positive perception of our revised manuscript. Please find below our replies to the remaining important remarks by reviewer 1. As before, all changes to the text are in red font.

Reviewer #1 (Remarks to the Author):

The authors made a great effort in revising the manuscript and added important new data, but several key issues remain:

1-The specificity of the epithelial injury states is overstated, as the so-called “TCMR-associated” states are also enriched in ABMR, mixed rejection, AKI, and CKD. These should be reframed as general maladaptive epithelial programs with prognostic rather than diagnostic value.

Reply: We thank the reviewer for the positive assessment of our revised manuscript and for this important remark. While we fully agree with the reviewer’s perspective, we would like to emphasize two points: 1. It is indeed very likely that the observed epithelial cell states represent general maladaptive programs. However, conclusively proving that these states are truly identical across different conditions is a non-trivial task that would require additional studies beyond the scope of this manuscript. Therefore, we prefer to remain cautious in making this claim. 2. By using the terms TCMR-associated or TCMR-induced cell states, we did not intend to imply TCMR-specific. Our mouse data indeed indicate that these cell states can be induced by TCMR, but we fully acknowledge that they may also arise under other conditions.

We agree that our previous wording could potentially be misleading and have therefore revised the terminology throughout the manuscript to clarify that these states are induced during TCMR but are likely not specific to it. We also added corresponding clarifications to the discussion. We hope the reviewer finds the revised wording more accurate and appropriate.

2-The issue of causality is not established, since the study relies on transcriptional correlations and spatial proximity analyses. Claims of therapeutic targetability without functional perturbation or mechanistic studies this remains speculative and therefore s should be tempered.

Reply: We thank the reviewer for this important remark. We fully agree with this assessment and have accordingly toned down or removed the respective statements from the manuscript.

3- The limitations of bulk transcriptomic analyses remain, because prognostic associations are presented without multivariable adjustment or controls for cell composition, which undermines the independence of the signal.

Reply: We thank the reviewer for this important remark. We fully agree that a definitive statistical assessment would ideally include multivariable adjustment with additional clinical covariates to control for potential confounders, as well as formal correction for cell composition. However, detailed clinical information was not available for all samples, and the number of TCMR biopsies (n = 95) is likely underpowered for a robust multivariable model. We have now explicitly acknowledged this limitation in the discussion and emphasize that the reported findings should be interpreted as associative rather than fully independent prognostic effects.

4- The human snRNA-seq dataset is underpowered, with only three TCMR and three stable

biopsies. Heterogeneity within PT-h2 (e.g., KIM1/VCAM1 vs. STK39/ACSM3 expression) remains unresolved.

Reply: Thank you for this important remark. Indeed, it is very likely that additional clusters (including a potential subdivision of the current PT Injury h2 cluster) may emerge with an increased number of human snRNA-seq samples. We attempted to transparently address this potential heterogeneity by immunofluorescence staining (Fig. 7A). Nevertheless, Fig. 6C shows that most top marker genes of PT Injury h2 overlap with those of PT Injury m4, and Fig. 6E further demonstrates the strongest correspondence between these two clusters. Based on these observations, we consider it most likely that a substantial part of the PT Injury m4 gene expression profile is indeed represented by PT Injury h2.

We fully agree, however, that the full complexity of PT subclustering in human TCMR may not be captured with our current approach. We have therefore incorporated this important point as a limitation and included respective statements in the discussion.

5- The ambiguity of the TAL-h1 state persists, as it is prognostically relevant but paradoxically depleted in TCMR and lacks a clear mouse counterpart. Its biological meaning requires clarification.

Reply: We thank the reviewer for highlighting this important point. As correctly noted, TAL Injury h1 is associated with adverse graft outcome but is relatively depleted in TCMR and lacks a clear mouse equivalent. In the revised discussion, we now elaborate on this apparent paradox. We propose that TAL Injury h1 may represent a general stress- or recovery-associated TAL program independent of alloimmune injury, potentially linked to hemodynamic or metabolic stress. In addition, we acknowledge that its lower relative abundance in TCMR could reflect proportional effects in the deconvolution analysis, where the expansion of more severely injured TAL Injury h2 states reduces the relative contribution of TAL Injury h1. We therefore consider TAL Injury h1 a non-TCMR-induced injury phenotype with adverse prognostic implications that warrants further investigation.

6- The claims about persistence of injury states are insufficiently supported, as they are based on only 12 patients with follow-up biopsies. These findings should be described as exploratory rather than definitive.

Reply: Thank you for this comment. We now explicitly mention this as an exploratory effort with a limited sample size throughout the manuscript.